Letter

# Large-scale genome-wide association analyses identify novel genetic loci and mechanisms in hypertrophic cardiomyopathy

Hypertrophic cardiomyopathy (HCM) is an important cause of morbidity and mortality with both monogenic and polygenic components. Here, we report results from a large genome-wide association study and multitrait analysis including 5,900 HCM cases, 68,359 controls and 36,083 UK Biobank participants with cardiac magnetic resonance imaging. We identified 70 loci (50 novel) associated with HCM and 62 loci (20 novel) associated with relevant left ventricular traits. Among the prioritized genes in the HCM loci, we identify a novel HCM disease gene, *SVIL*, which encodes the actin-binding protein supervillin, showing that rare truncating *SVIL* variants confer a roughly tenfold increased risk of HCM. Mendelian randomization analyses support a causal role of increased left ventricular contractility in both obstructive and nonobstructive forms of HCM, suggesting common disease mechanisms and anticipating shared response to therapy. Taken together, these findings increase our understanding of the genetic basis of HCM, with potential implications for disease management.

HCM is a disease of the cardiac muscle characterized by thickening of the left ventricular (LV) wall with or without obstruction of flow (obstructive, oHCM; nonobstructive, nHCM). HCM is associated with an increased risk of arrhythmia, heart failure, stroke and sudden death. Previously viewed as a Mendelian disease with rare pathogenic variants in cardiac sarcomere genes identified in ~35% of cases (HCM$_{SARC+}$), HCM is now known to have complex and diverse genetic architectures[1]. Previous studies have established that common genetic variants underlie a large portion of disease heritability in HCM not caused by rare pathogenic variants (sarcomere-negative (HCM$_{SARC-}$)) and partly explain the variable expressivity in HCM patients carrying pathogenic variants (sarcomere-positive (HCM$_{SARC+}$)), but such studies had limited power to identify a large number of significant loci[2,3].

We report a meta-analysis of seven case–control HCM genome-wide association study (GWAS) datasets comprising a total of 5,900 HCM cases, 68,359 controls and 9,492,702 variants with a minor allele frequency (MAF) > 1% (Supplementary Table 1; see study flowchart in Fig. 1). We identified 34 loci significantly associated with HCM (at $P < 5 \times 10^{-8}$), of which 15 were novel (Table 1 and Supplementary Figs. 1 and 2a). Stratified analyses in HCM$_{SARC+}$ (1,776 cases) and HCM$_{SARC-}$ (3,860 cases) identified a further five loci (Table 1, Supplementary Fig. 2b and Supplementary Table 2). Using conditional analysis[4], we identified more independent associations with HCM, HCM$_{SARC+}$, and HCM$_{SARC-}$ with a false discovery rate (FDR) < 1% (Supplementary Table 3). We estimated the heritability of HCM attributable to common genetic variation ($h^2_{SNP}$) in the all-comer analysis to be 0.17 ± 0.02 using linkage disequilibrium (LD) score regression (LDSC)[5], and found higher estimates (0.25 ± 0.02) using genome-based restricted maximum likelihood (GREML)[6], with higher $h^2_{SNP}$ in HCM$_{SARC-}$ (0.29 ± 0.02) compared with HCM$_{SARC+}$ (0.16 ± 0.04) (Supplementary Table 4).

To further maximize HCM locus discovery, we performed a multitrait analysis of GWAS (MTAG)[7] (Fig. 2). We first completed a GWAS

e-mail: rafik.tadros@umontreal.ca; j.ware@imperial.ac.uk; c.r.bezzina@amsterdamumc.nl; hugh.watkins@rdm.ox.ac.uk

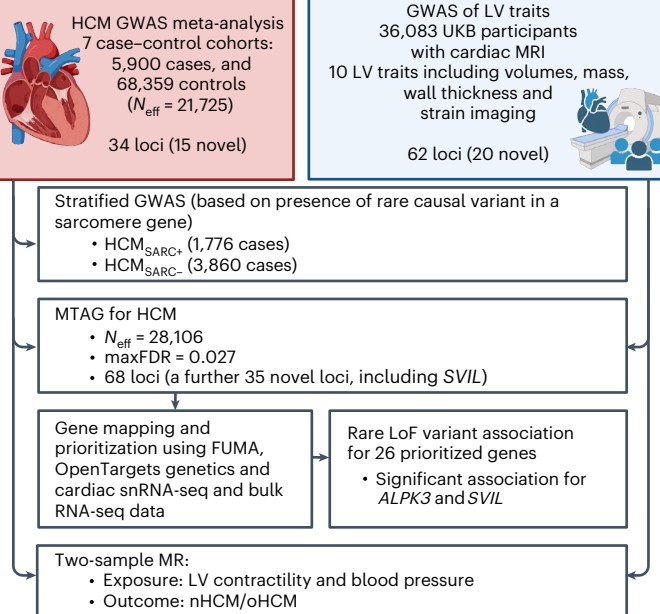

**Fig. 1 | Study flowchart.** Flowchart of meta-analysis of seven case–control HCM GWAS datasets, GWAS of LV traits and downstream analyses. Created using BioRender.com.

of ten cardiomyopathy-relevant LV traits in 36,083 participants of the UK Biobank (UKB), using machine learning assessment of cardiac magnetic resonance (CMR) imaging[8] for LV volumes, wall thickness and myocardial strain (Supplementary Table 5 and Supplementary Figs. 3–12). We discovered 62 loci associated with LV traits (20 novel) (Supplementary Table 6). LDSC analyses[9] demonstrated high genetic correlations (rg) between LV traits within three clusters (contractility, volume and mass) and with HCM (Fig. 2 and Supplementary Table 7). Leveraging such correlations, we performed an HCM MTAG by including the three LV traits most correlated with HCM (one trait from each cluster), namely global circumferential strain (contractility cluster; rg = −0.62), LV end-systolic volume (volume cluster; rg = −0.48) and the ratio of LV mass to end-diastolic volume (mass cluster; rg = 0.63). MTAG resulted in a substantial increase in mean $\chi^2$ equivalent to an increase in effective sample size ($N_{eff}$) of the HCM GWAS of ~29% (from 21,725 to 28,106), with an estimated upper bound of the false discovery rate (maxFDR)[7] of 0.027. Effect estimates derived from MTAG were strongly correlated with those from GWAS (Supplementary Fig. 13). MTAG resulted in a substantial step up in loci discovered, identifying a total of 68 loci associated with HCM at $P < 5 \times 10^{-8}$, including 48 that have not been published previously (13 novel loci also identified in the single-trait HCM GWAS, and 35 additional novel loci from MTAG) (Fig. 3, Supplementary Table 8 and Supplementary Fig. 14). Two of the 34 loci reaching genome-wide significance in the HCM GWAS were not significant in MTAG (loci mapped to *TRDN/HEY2* and *CHPF*). The total number of loci identified in GWAS or MTAG is therefore 70 (50 novel). Although it was not possible to test for replication for the 35 novel MTAG loci, a previous study strongly supports the robustness of the HCM-LV traits MTAG approach, whereby all ten HCM loci uncovered using MTAG in this previous study were independently validated[3], and all reach $P < 5 \times 10^{-8}$ in the present GWAS.

MAGMA[10] gene set analysis identified several significant gene sets linked to muscle, contractility and sarcomeric function (Supplementary Table 9), and tissue expression analysis pointed to cardiac tissue (LV and atrial appendage (AA)) (Supplementary Table 10). Within cardiac tissue, we further explored the contribution of specific cell types in HCM by leveraging available single-nuclei RNA sequencing (snRNA-seq) data from donor human hearts[11]. Using sc-linker[12], we

identified significant enrichment of heritability in cardiomyocyte and adipocyte cell types (cardiomyocyte, FDR-adjusted $P = 1.8 \times 10^{-6}$; adipocyte, FDR-adjusted $P = 3.0 \times 10^{-3}$) and cell states (Supplementary Fig. 15).

Lead variants at GWAS and MTAG loci map to noncoding sequences of the genome, with only a few exceptions that are missense variants in *BAG3*, *ADPRHL1*, *PROB1* and *RNF207* (Table 1 and Supplementary Tables 8 and 11). Prioritization of potential causal genes in HCM MTAG loci was performed using OpenTargets variant-to-gene (V2G) mapping[13] (Supplementary Table 12) and FUMA[14] (Supplementary Table 13). Of all prioritized genes, 26 were selected based on concordance in both OpenTargets (top three genes per locus) and FUMA, as well as LV-specific expression in bulk RNA-seq data (genotype-tissue expression project (GTEx) v.8) and expression in cardiomyocytes using publicly available snRNA-seq data[15] (Fig. 4a and Supplementary Tables 12 and 13). Of those 26 genes, 14 are in novel loci and include genes involved in cardiomyocyte energetics and metabolism (*RNF207* (ref. 16), *MLIP*[17]), myocyte differentiation and transcriptional regulation (*MITF*[18], *PROX1* (ref. 19), *TMEM182* (ref. 20)), myofibril assembly (*SVIL*[21]) and calcium handling and contractility (*PDE3A*[22], *SRL*[23]). To identify further genes associated with HCM, we performed a transcriptome-wide association study (TWAS) using S-MultiXcan[24] with the MTAG summary statistics and cardiac tissues (LV and AA) from GTEx v.8. TWAS identified 127 genes significantly associated with HCM at $P < 3.7 \times 10^{-6}$ (Supplementary Table 14), of which 50 were not mapped to MTAG loci using either FUMA or OpenTargets, including *HHATL* ($P = 1 \times 10^{-11}$)—a gene of uncertain function prioritized based on dominant LV expression and whose depletion in zebrafish may lead to cardiac hypertrophy[25]. Finally, we used OpenTargets to explore association of the 70 lead single nucleotide polymorphisms (SNPs) (or any other SNP in linkage disequilibrium, $r^2 > 0.5$) with published cardiovascular, metabolic or other traits (Supplementary Table 15). Of the 70 loci associated with HCM, 51 were previously associated at $P < 5 \times 10^{-8}$ with cardiovascular and/or cardiometabolic traits, including ECG measures, body mass, blood pressure, atrial fibrillation, left ventricular structure/function, atherosclerotic cardiovascular disease and lipids.

GWAS loci often colocalize with genes harboring disease-causing rare variants[26]. To identify novel HCM disease genes, we explored whether rare (MAF < 10^−4) predicted loss-of-function (LoF) variants in the 26 prioritized genes from significant GWAS/MTAG loci are associated with HCM. We performed case–control burden testing using sequencing data from BioResource of Rare Diseases (BRRD), Genomics England (GEL), UKB and the Oxford Medical Genetics Laboratory (OMGL; only for *SVIL*), followed by a fixed-effects model IVW meta-analysis comprised of up to 2,502 unrelated HCM cases and 486,217 controls (Fig. 4b and Supplementary Table 16a). Rare LoF variants in *ALPK3* and *SVIL* were significantly associated with HCM at the Bonferroni-corrected $P < 0.0019$ (0.05/26). While truncating variants in *ALPK3* have previously been shown to cause HCM and are now included in most clinical testing panels[26–28], *SVIL* represents a novel HCM gene with a comparable effect size for LoF variants (odds ratio (OR), 10.5,; 95% confidence intervals (CI), 4.3–26.1; $P = 3.6 \times 10^{-7}$). As exploratory analyses, we also performed exome-wide gene-based burden testing for LoF variants using two MAF filters (< 10^−3 and <10^−4) and report the summary statistics in the supplement (Supplementary Tables 17 and 18 and Supplementary Figs. 16–18). Effect estimates for rare *SVIL* LoF variants do not show significant heterogeneity across the four datasets (Supplementary Fig. 19), and the associations remain significant when excluding each dataset one at a time (Supplementary Table 16b). Furthermore, synonymous variant burden testing was performed as a negative control and did not show significant associations (Fig. 4b and Supplementary Table 16c). *SVIL* LoF variants found in eight unrelated cases are listed in Supplementary Table 19 and Fig. 4c. None of the eight unrelated HCM cases that carry a *SVIL* LoF variant carries any other pathogenic or likely pathogenic variant. Family screening

## Table 1 | Lead variants from the HCM GWAS

| Lead SNP | GRCh37 | EA/ NEA | EAF | OR (95% CI) | P | Q | Locus name | GWS in HCM$_{SARC+}$ | GWS in HCM$_{SARC-}$ |
|---|---|---|---|---|---|---|---|---|---|
| **Genome-wide significant loci from all HCM meta-analysis** | | | | | | | | | |
| rs2234962 | 10:121429633 | C/T | 0.21 | 1.45 (1.38–1.52) | $1.39\times10^{-49}$ | 0.36 | *BAG3* (missense) | • | • |
| rs2644262 | 18:34223566 | C/T | 0.29 | 1.38 (1.32–1.45) | $1.79\times10^{-43}$ | 0.43 | *FHOD3/TPGS2* | • | • |
| rs78310129 | 11:56793878 | T/C | 0.01 | 3.53 (2.92–4.27) | $9.79\times10^{-39}$ | 1.59E-09 | *MYBPC3* | • | |
| rs1048302 | 1:16340879 | T/G | 0.33 | 1.28 (1.23–1.34) | $8.47\times10^{-30}$ | 0.59 | *HSPB7* | | • |
| rs2070458* | 22:24159307 | A/T | 0.22 | 1.30 (1.24–1.37) | $5.93\times10^{-25}$ | 0.09 | *VPREB3/SMARCB1* | | • |
| rs3176326* | 6:36647289 | A/G | 0.21 | 1.30 (1.24–1.37) | $3.18\times10^{-24}$ | 0.14 | *CDKN1A* | | • |
| rs12212795 | 6:118654308 | C/G | 0.05 | 1.51 (1.39–1.65) | $4.76\times10^{-22}$ | 0.33 | *SLC35F1/PLN* | | • |
| rs4577128* | 17:64308473 | C/T | 0.57 | 1.23 (1.18–1.29) | $3.26\times10^{-21}$ | 0.97 | *PRKCA* | | • |
| rs393838 | 17:43705756 | C/G | 0.23 | 1.26 (1.20–1.32) | $5.02\times10^{-21}$ | 0.91 | *CRHR1/MAPT* | | • |
| rs8033459* | 15:85253258 | T/C | 0.46 | 1.20 (1.15–1.25) | $7.04\times10^{-18}$ | 0.66 | *ALPK3/NMB* | • | • |
| rs11196085* | 10:114505037 | C/T | 0.28 | 1.22 (1.16–1.28) | $1.85\times10^{-17}$ | 0.75 | *VTI1A/TCF7L2* | | • |
| rs7301677 | 12:115381147 | C/T | 0.74 | 1.22 (1.16–1.29) | $7.01\times10^{-16}$ | 0.27 | *TBX3* | | • |
| rs2177843* | 10:75409877 | T/C | 0.16 | 1.26 (1.19–1.34) | $2.80\times10^{-15}$ | 0.91 | *MYOZ1/SYNPO2L* | | • |
| rs41306688 | 13:114078558 | C/A | 0.03 | 1.60 (1.42–1.80) | $3.04\times10^{-15}$ | 0.22 | *ADPRHL1* (missense) | | • |
| rs2191445* | 5:57011469 | T/A | 0.80 | 1.23 (1.17–1.30) | $8.22\times10^{-14}$ | 0.37 | *ACTBL2* | | • |
| rs4894803* | 3:171800256 | G/A | 0.41 | 1.18 (1.13–1.24) | $2.19\times10^{-13}$ | 0.63 | *FNDC3B* | | • |
| rs13061705 | 3:14291129 | C/T | 0.69 | 1.19 (1.13–1.25) | $5.67\times10^{-13}$ | 0.68 | *SLC6A6/LSM3* | | • |
| **rs13021775** | **2:37059557** | **C/G** | **0.50** | **1.17 (1.12–1.23)** | **$5.98\times10^{-13}$** | 0.50 | ***STRN*** | | • |
| **rs8006225** | **14:95219657** | **G/T** | **0.83** | **1.22 (1.15–1.30)** | **$2.64\times10^{-11}$** | 0.15 | ***GSC*** | | • |
| rs10052399* | 5:138668504 | T/C | 0.27 | 1.18 (1.12–1.24) | $3.99\times10^{-11}$ | 0.03 | *SPATA24* | | |
| rs66520020* | 7:128438284 | T/C | 0.16 | 1.21 (1.14–1.28) | $5.87\times10^{-11}$ | 0.96 | *CCDC136/FLNC* | | |
| **rs12460541** | **19:46312077** | **G/A** | **0.66** | **1.16 (1.11–1.21)** | **$6.01\times10^{-11}$** | 0.13 | ***DMPK/SYMPK*** | | |
| **rs7461129** | **8:125861374** | **T/C** | **0.31** | **1.16 (1.11–1.21)** | **$8.19\times10^{-11}$** | 0.87 | ***MTSS1*** | | |
| **rs56005624** | **2:179774634** | **G/T** | **0.14** | **1.21 (1.14–1.28)** | **$8.31\times10^{-11}$** | 0.62 | ***CCDC141/SESTD1*** | | • |
| **rs7824244** | **8:21802432** | **A/G** | **0.14** | **1.22 (1.14–1.29)** | **$2.39\times10^{-10}$** | 0.34 | ***XPO7*** | • | |
| **rs12270374** | **11:14375079** | **C/T** | **0.36** | **1.14 (1.09–1.20)** | **$6.85\times10^{-10}$** | 0.92 | ***RRAS2/COPB1*** | | |
| **rs62222424** | **21:30530131** | **G/A** | **0.93** | **1.32 (1.20–1.44)** | **$1.21\times10^{-9}$** | 0.69 | ***CCT8*** | | |
| **rs11687178** | **2:11584197** | **C/A** | **0.65** | **1.14 (1.09–1.19)** | **$7.70\times10^{-9}$** | 0.26 | ***E2F6/ROCK2*** | | |

## Table 1 (continued) | Lead variants from the HCM GWAS

| Lead SNP | GRCh37 | EA/NEA | EAF | OR (95% CI) | *P* | *Q* | Locus name | GWS in HCM$_{SARC+}$ | GWS in HCM$_{SARC-}$ |
|---|---|---|---|---|---|---|---|---|---|
| **rs9320939** | **6:123818871** | **A/G** | **0.49** | **1.13 (1.08–1.18)** | **1.04×10⁻⁸** | 0.04 | **TRDN/HEY2** | | • |
| **rs2540277** | **2:103426177** | **C/T** | **0.94** | **1.32 (1.19–1.45)** | **2.31×10⁻⁸** | 0.84 | **TMEM182/MFSD9** | | |
| **rs6566955** | **18:55922789** | **G/A** | **0.31** | **1.14 (1.08–1.19)** | **2.93×10⁻⁸** | 0.16 | **NEDD4L** | | |
| **rs13004994** | **2:220406239** | **T/G** | **0.46** | **1.13 (1.08–1.18)** | **3.02×10⁻⁸** | 0.12 | **CHPF** | | |
| **rs2645210** | **10:4098453** | **A/G** | **0.19** | **1.16 (1.10–1.23)** | **3.94×10⁻⁸** | 0.52 | **KLF6/AKR1E2** | | |
| **rs113907726** | **14:53316867** | **G/T** | **0.19** | **1.16 (1.10–1.22)** | **4.10×10⁻⁸** | 0.27 | **FERMT2/ERO1A** | | |
| **Additional loci discovered in HCM$_{SARC+}$ or HCM$_{SARC-}$** | | | | | | | | | |
| rs9311485 | 3:52987645 | T/G | 0.25 | 1.13 (1.08–1.19) | 1.86×10⁻⁷ | 0.09 | ITIH3/SFMBT1 | | • |
| rs77963625 | 12:46446897 | C/T | 0.03 | 1.38 (1.22–1.57) | 2.97×10⁻⁷ | 0.24 | SCAF11 | | • |
| rs846111 | 1:6279370 | G/C | 0.73 | 1.14 (1.08–1.20) | 6.32×10⁻⁷ | 0.52 | RNF207 (missense) | | • |
| rs58747679 | 12:26348304 | T/C | 0.71 | 1.12 (1.07–1.18) | 1.30×10⁻⁶ | 0.15 | SSPN | | • |
| rs112787369 | 14:68252852 | T/A | 0.04 | 1.21 (1.08–1.35) | 6.04×10⁻⁴ | 0.62 | ZYVE26 (missense) | • | |

All reported summary statistics refer to the all HCM case–control meta-analysis results, including for loci identified only in the HCM$_{SARC+}$ and HCM$_{SARC-}$ stratified analyses. The table is sorted in increasing order of the all-comer *P* values. Novel loci are shown in bold. An asterisk marks loci that reached significance in a previous multitrait analysis of GWAS (MTAG)[3] and now reach significance in the present GWAS. Locus naming was performed primarily by OpenTargets[13], also considering functional mapping and annotation of GWAS (FUMA)[14] mapping, and previous rare variant associations with HCM[26]. Dots indicate the presence of GWS. EA/NEA, effect and noneffect alleles; EAF, effect allele frequency; GWS, genome-wide significance ($P \leq 5 \times 10^{-8}$); *Q*, Cochrane's heterogeneity test *P* value.

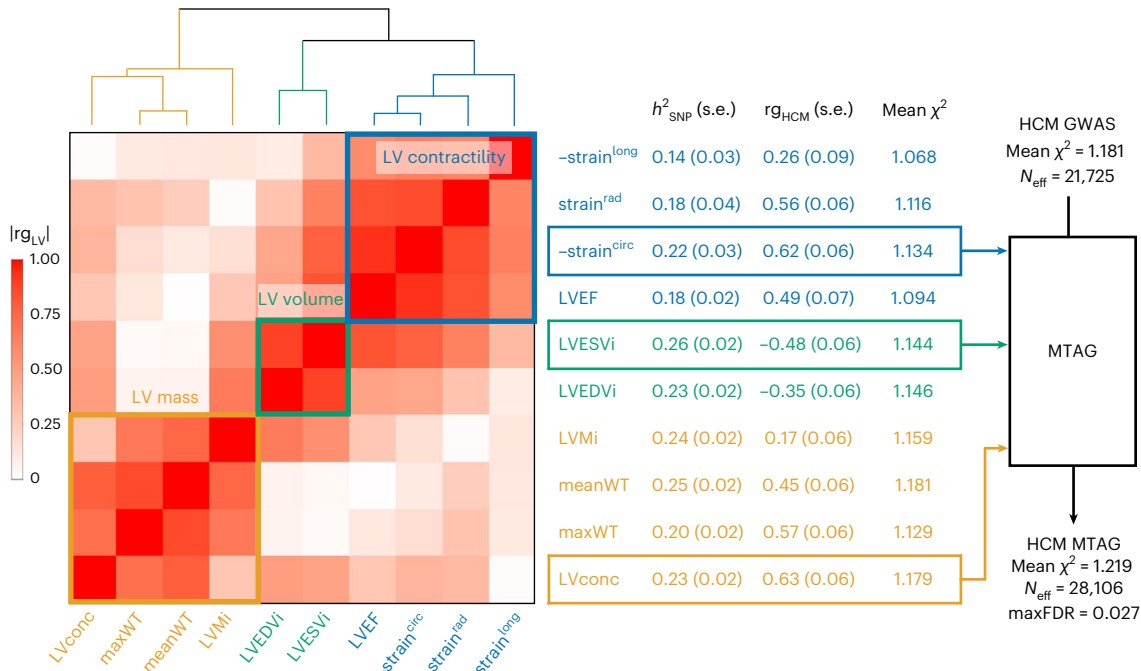

**Fig. 2 | Genetic correlation of LV traits and HCM and use of MTAG to empower locus discovery.** Pairwise genetic correlation between LV traits shown in heatmap as absolute values ($|rg_{LV}|$) ranging from 0 (white) to 1 (red). LV traits are sorted into three clusters based on $|rg_{LV}|$ along the *x* and *y* axes using Euclidean distance and complete hierarchical clustering: LV contractility (blue), volume (bluish green) and mass (orange) (see dendrogram on top). The table in the middle shows the individual LV trait $h^2_{SNP}$ and genetic correlation with HCM ($rg_{HCM}$), with corresponding s.e. The trait with the strongest correlation (based on $rg_{HCM}$) in each of the three clusters was carried forward for MTAG to empower locus discovery in HCM. MTAG resulted in an increase in $N_{eff}$, based on number of cases and controls and increase in mean $\chi^2$ statistic from 21,725 to 28,106, with an estimated maxFDR of 0.027. Since strain$^{circ}$ and strain$^{long}$ are negative values where increasingly negative values reflect increased contractility, we show −strain$^{circ}$ and −strain$^{long}$ to facilitate interpretation of $rg_{HCM}$ sign. Full $rg_{LV}$ and $rg_{HCM}$ results are shown in Supplementary Table 7.

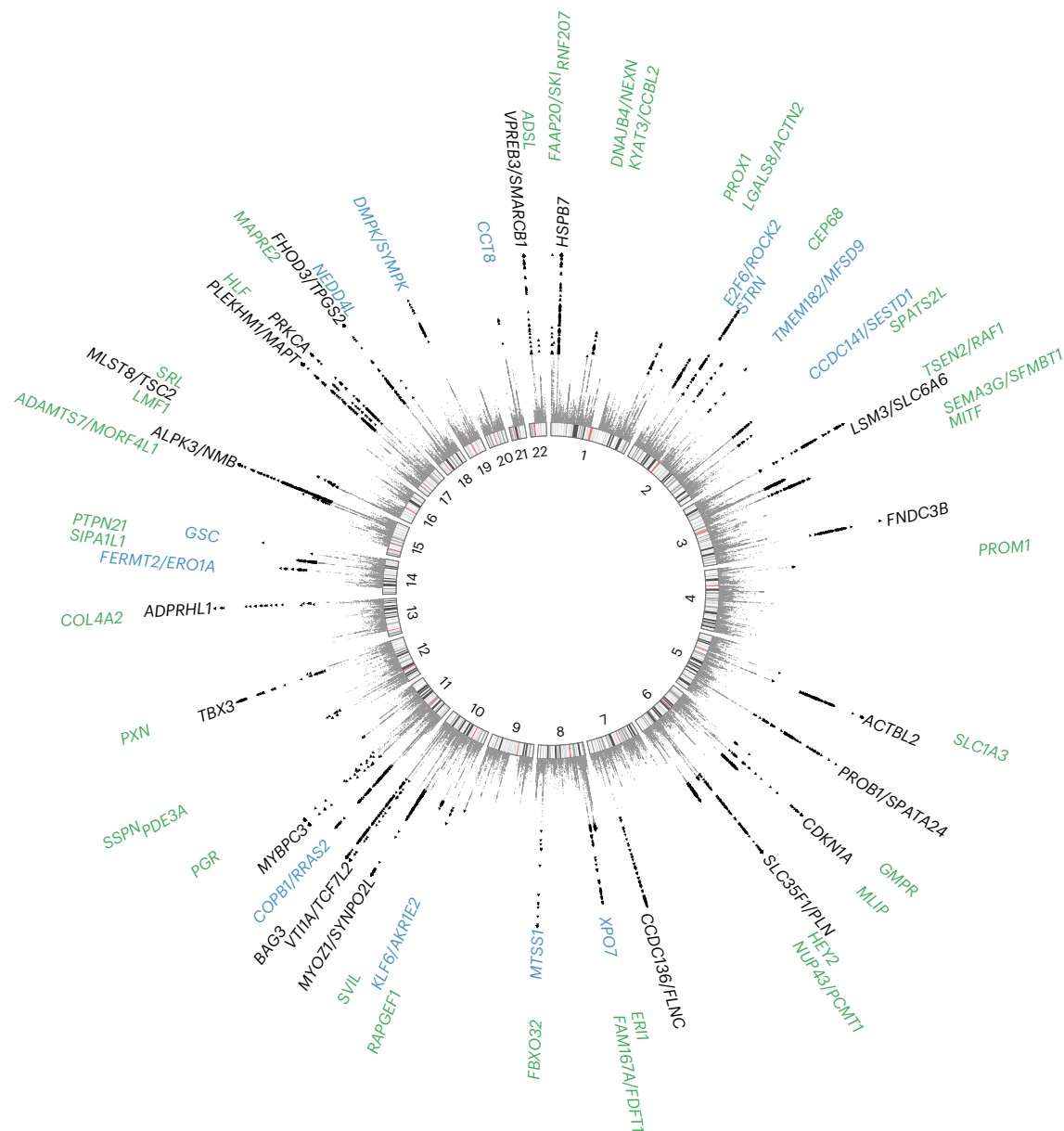

**Fig. 3 | Circular Manhattan plot of HCM summary statistics from MTAG analysis.** Previously published loci are identified in black (*n* = 20), novel loci discovered by single-trait all-comer GWAS meta-analysis are identified in blue (*n* = 13) and other novel loci from MTAG are identified in green (*n* = 35). Two other loci reaching GWAS significance threshold in the single-trait HCM GWAS meta-analysis but not reaching significance in MTAG are not shown (mapped to *TRDN*/*HEY2* and *CHPF*; Table 1). *P* values are not corrected for multiple testing and correspond to the HCM MTAG including the fixed-effects meta-analysis of seven HCM case–control GWAS and three LV traits (Fig. 2). Significant variants with $P < 5 \times 10^{-8}$ are shown as black triangles. Results with $P < 1 \times 10^{-15}$ are assigned $P = 1 \times 10^{-15}$. Locus naming was performed primarily by OpenTargets gene prioritization considering FUMA and previous gene association with Mendelian HCM. See Supplementary Table 8 for loci details.

provided limited evidence of cosegregation. In one family, variant *SVIL*:p.(Gln255Ter) was carried by two cousins with HCM and, in another family, variant *SVIL*:p.(Arg1616Ter) was carried by two siblings with HCM. *SVIL* encodes supervillin, a large, multidomain actin and myosin binding protein with several muscle and nonmuscle isoforms, of which the muscle isoform has known roles in myofibril assembly and Z-disk attachment[21]. SVIL is highly expressed in cardiac, skeletal and smooth muscle myocytes in the GTEx v.9 snRNA-seq dataset[29], and *SVIL* morpholino knockdown in zebrafish produces cardiac abnormalities[30]. In humans, LoF *SVIL* variants have been associated with smaller descending aortic diameter[31], and homozygous LoF *SVIL* variants have been shown to cause a skeletal myopathy with mild cardiac features (left ventricular hypertrophy)[32]. Of interest, common variants in the *SVIL* locus are also associated with dilated cardiomyopathy (DCM)[3] and,

using a Bayesian pairwise analysis approach (GWAS-PW[33]) including the present HCM GWAS meta-analysis and a published DCM GWAS[34], we show that DCM and HCM share the same causal SNP but with the expected opposite directions of effect (Supplementary Table 20). Taken together, these data support *SVIL* as the likely causal gene in the HCM GWAS locus and identify *SVIL* as a novel disease gene for HCM, in which rare LoF alleles have an effect size similar to that of minor HCM disease genes tested in clinical practice[26–28].

Rare sarcomeric gene variants that cause HCM have been shown to result in increased contractility, and cardiac myosin inhibitors attenuate the development of sarcomeric HCM in animal models[35]. Previous data from GWAS and Mendelian randomization (MR) also support a causal association of increased LV contractility with HCM, extending beyond rare sarcomeric variants[3]. Pharmacologic modulation of LV contractility

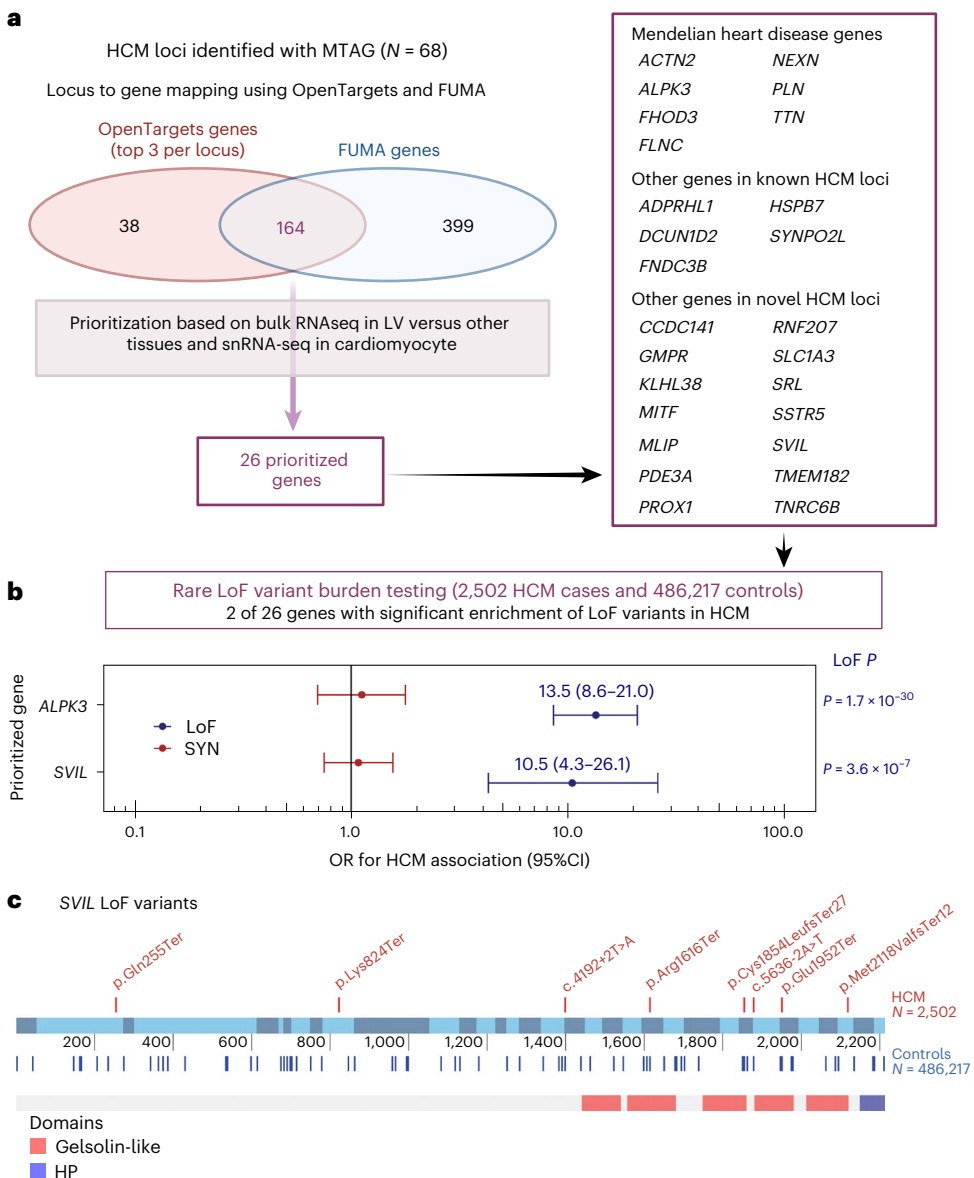

**Fig. 4 | HCM locus-to-gene mapping, prioritization and rare LoF association testing identifies *SVIL* as a new HCM disease gene. a**, HCM locus-to-gene mapping and prioritization based on cardiac expression. Locus-to-gene mapping was done using the OpenTargets[13] V2G pipeline (release of 12 October 2022) for all 68 lead variants at the HCM MTAG loci and using FUMA[14] for the HCM MTAG summary statistics (see Methods for detailed parameters). Of 164 genes mapped using both FUMA and OpenTargets (top 3 genes per locus), 26 were prioritized because of either high specificity of LV expression using the bulk RNA-seq data of the GTEx project[38] release v.8 and/or high expression in cardiomyocytes using snRNA-seq data[15]. See Methods and Supplementary Tables 12 and 13 for details. **b**, Rare (MAF < 10[−4]) LoF variant association analyses with HCM versus controls performed for all 26 genes using sequencing data in up to 2,502 unrelated HCM

cases and 486,217 controls from four datasets followed by IVW meta-analysis. Association of rare synonymous (SYN) variants was also performed as a negative control. Results shown restricted to two genes (*ALPK3* and *SVIL*) reaching the Bonferroni-corrected threshold of *P* < 0.0019 (0.05/26) in the IVW meta-analysis. Filled circles and error bars represent the OR and their 95% CI, respectively, from the meta-analysis for LoF (blue) and SYN (red). *P* values shown are not corrected for multiple testing. Full results appear in Supplementary Table 16. **c**, Schematic of the rare LoF *SVIL* variants in HCM cases (top) and controls (bottom) along the linear structure of *SVIL*, showing the Gelsolin-like and headpiece (HP) domains. The coordinates reflect the codon numbers, and the colored bars are the exons. Detailed variant annotation appears in Supplementary Table 19. Panel **a** was created using BioRender.com.

using myosin inhibitors has been approved recently in the treatment of HCM associated with LV obstruction (oHCM)[36,37], but remains of uncertain utility in nHCM where no specific therapy currently exists. To further dissect the specific implication of LV contractility in nHCM and oHCM, we performed two-sample MR, testing the causal association of LV contractility as exposure, with HCM, nHCM and oHCM as outcomes. GWAS of nHCM (2,491 cases) and oHCM (964 cases) were performed (Supplementary Table 2), showing substantially shared genetic basis between nHCM and oHCM (rg = 0.87; s.e. 0.13; *P* = 4 × 10[−11]) (Supplementary Table 8). LV contractility in the general population was assessed with CMR using a

volumetric method (LV ejection fraction (LVEF)), and three-dimensional tissue deformation methods (that is, global LV strain in the longitudinal (strain[long]), circumferential (strain[circ]) and radial (strain[rad]) directions). Results from the primary MR inverse variance weighted (IVW) analysis are shown in Fig. 5a, and sensitivity analyses results appear in Supplementary Table 21 and Supplementary Figs. 20 and 21. Although significant heterogeneity in the exposure–outcome effects and potential violations of MR assumptions are possible limitations, MR findings support a causal association between increased LV contractility and increased risk for both nHCM and oHCM, with a substantial risk increase of 12-fold and

**a**

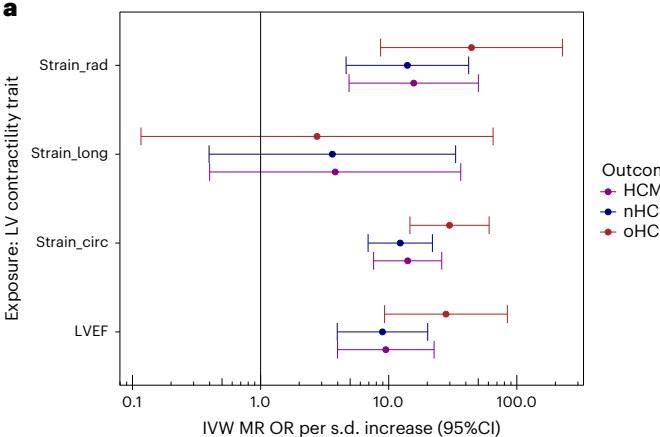

**b**

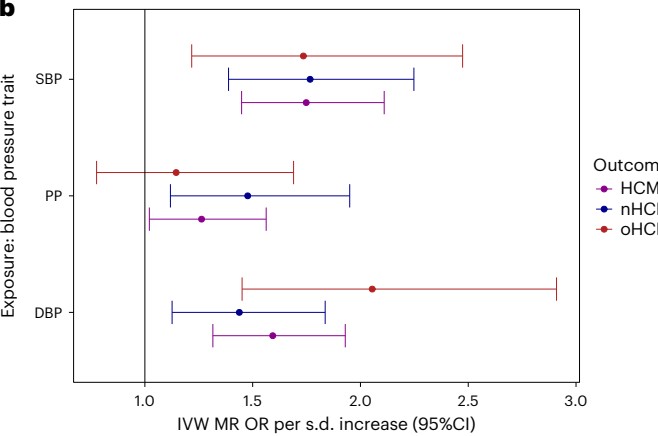

**Fig. 5 | MR analysis of LV contractility and blood pressure on risk of oHCM and nHCM.** In both panels, filled circles represent the OR per s.d. increase inferred from the IVW two-sample MR. Error bars represent the 95% CI of the OR. **a**, MR suggests causal association of LV contractility (exposure) with HCM, oHCM and nHCM (outcomes), where increased contractility increases disease risk. Genetic instruments for LV contractility were selected from the present GWAS of LVEF and LV strain in the radial (strain_rad), longitudinal (strain_long) and circumferential (strain_circ) directions in 36,083 participants of the UKB without cardiomyopathy and with available CMR. To facilitate interpretation of effect directions, OR for strain_circ and strain_long reflect those of increased contractility (more negative strain_circ and strain_long values). The outcome HCM GWAS included 5,900 HCM cases versus 68,359 controls. Of those, 964 cases and 27,163 controls were included in the oHCM GWAS and 2,491 cases and 27,109 were included in the nHCM GWAS. Note a logarithmic scale in the *x* axis. **b**, MR suggests causal associations of SBP and DBP with HCM, nHCM and oHCM. Genetic instruments for SBP, DBP and PP (SBP − DBP) were selected from a published GWAS including up to 801,644 people[39]. See Supplementary Table 21 for full MR results.

29-fold per s.d. increase in the absolute value of strain[circ], respectively (Fig. 5a). These data suggest that increased contractility is involved in both oHCM and nHCM development, and thus myosin inhibitors currently approved for symptom control in oHCM may also be of clinical benefit in nHCM. Finally, we also performed MR analyses exploring whether increased systolic (SBP) and diastolic (DBP) blood pressure, and pulse pressure (PP = SBP − DBP) are causally associated with nHCM and oHCM. As for LV contractility, the causal association of SBP and DBP with HCM[2] extended to both oHCM and nHCM subgroups (Fig. 5b, Supplementary Table 21 and Supplementary Fig. 22), suggesting that lowering blood pressure may be a therapeutic target to mitigate disease progression for both nHCM and oHCM.

The large number of new susceptibility loci arising from this work support new inferences regarding disease mechanisms in HCM. With

the identification of the role of *SVIL*, we have uncovered further evidence that a subset of genes underlies both monogenic and polygenic forms of the condition. However, this shared genetic architecture does not extend to the core sarcomere genes that cause monogenic HCM; instead, the common variant loci implicate processes outside the myofilament, thereby widening our biological understanding. The shared mechanistic pathways between oHCM and nHCM suggest that the new class of myosin inhibitors may be effective in both settings, whereas the further exploration of newly implicated loci and pathways may in the future yield new treatment targets.

## Online content

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

Rafik Tadros [1,2,3,81]✉, Sean L. Zheng [4,5,6,81], Christopher Grace[7], Paloma Jordà [1,2], Catherine Francis[4,6], Dominique M. West[7], Sean J. Jurgens [3,8], Kate L. Thomson [7,9], Andrew R. Harper [7], Elizabeth Ormondroyd [7], Xiao Xu[5], Pantazis I. Theotokis [4,5,6], Rachel J. Buchan [4,5,6], Kathryn A. McGurk [4,5], Francesco Mazzarotto [4,10], Beatrice Boschi[11], Elisabetta Pelo[11], Michael Lee [4], Michela Noseda [4], Amanda Varnava[4,12], Alexa M. C. Vermeer[3,13,14], Roddy Walsh [3], Ahmad S. Amin [3,14,15], Marjon A. van Slegtenhorst[16], Nicole M. Roslin [17], Lisa J. Strug [17,18,19], Erika Salvi [20], Chiara Lanzani[21,22], Antonio de Marvao [5,23], Hypergenes InterOmics Collaborators*, Jason D. Roberts[24], Maxime Tremblay-Gravel[1,2], Genevieve Giraldeau[1,2], Julia Cadrin-Tourigny[1,2], Philippe L. L'Allier[1,2], Patrick Garceau[1,2], Mario Talajic[1,2], Sarah A. Gagliano Taliun [1,2], Yigal M. Pinto [3,14,15], Harry Rakowski[25], Antonis Pantazis[6], Wenjia Bai [26,27,28], John Baksi[4,6], Brian P. Halliday[4,6], Sanjay K. Prasad[4,6], Paul J. R. Barton [4,5,6], Declan P. O'Regan [5], Stuart A. Cook[5,29,30], Rudolf A. de Boer [31], Imke Christiaans [32], Michelle Michels[14,31], Christopher M. Kramer[33], Carolyn Y. Ho [34], Stefan Neubauer[35], HCMR Investigators*, Paul M. Matthews [27,36], Arthur A. M. Wilde[3,14,15,37], Jean-Claude Tardif [1,2], Iacopo Olivotto[38], Arnon Adler[25,39], Anuj Goel[7], James S. Ware [4,5,6,40,81]✉, Connie R. Bezzina [3,14,81]✉ & Hugh Watkins [7,81]✉

[1]Cardiovascular Genetics Centre and Research Centre, Montreal Heart Institute, Montreal, Quebec, Canada. [2]Faculty of Medicine, Université de Montréal, Montreal, Quebec, Canada. [3]Department of Experimental Cardiology, Amsterdam Cardiovascular Sciences, University of Amsterdam, Amsterdam UMC, Amsterdam, the Netherlands. [4]National Heart and Lung Institute, Imperial College London, London, UK. [5]MRC Laboratory of Medical Sciences, Imperial College London, London, UK. [6]Royal Brompton and Harefield Hospitals, Guy's and St. Thomas' NHS Foundation Trust, London, UK. [7]Division of

Cardiovascular Medicine, Radcliffe Department of Medicine, University of Oxford, John Radcliffe Hospital, Oxford, UK. [8]Cardiovascular Disease Initiative, Broad Institute of MIT and Harvard, Cambridge, MA, USA. [9]Oxford Genetics Laboratories, Churchill Hospital, Oxford, UK. [10]Department of Molecular and Translational Medicine, University of Brescia, Brescia, Italy. [11]Genetics Unit, Careggi University Hospital, Florence, Italy. [12]Imperial College Healthcare NHS Trust, Imperial College London, London, UK. [13]Department of Clinical Genetics, University of Amsterdam, Amsterdam UMC, Amsterdam, the Netherlands. [14]European Reference Network for Rare and Low Prevalence Complex Diseases of the Heart (ERN GUARD-HEART), Amsterdam, the Netherlands. [15]Department of Clinical Cardiology, Amsterdam Cardiovascular Sciences, University of Amsterdam, Amsterdam UMC, Amsterdam, the Netherlands. [16]Department of Clinical Genetics, Erasmus Medical Center, University Medical Center Rotterdam, Rotterdam, the Netherlands. [17]Program in Genetics and Genome Biology and The Centre for Applied Genomics, The Hospital for Sick Children, Toronto, Ontario, Canada. [18]Departments of Statistical Sciences and Computer Science, University of Toronto, Toronto, Ontario, Canada. [19]Division of Biostatistics, Dalla Lana School of Public Health, University of Toronto, Toronto, Ontario, Canada. [20]Neuroalgology Unit, Fondazione IRCCS Istituto Neurologico 'Carlo Besta', Milan, Italy. [21]Genomics of Renal Diseases and Hypertension Unit and Nephrology Operative Unit, IRCCS San Raffaele Hospital, Milan, Italy. [22]Vita-Salute San Raffaele University, Milan, Italy. [23]King's College London, London, UK. [24]Department of Medicine, Section of Cardiac Electrophysiology, Division of Cardiology, Western University, London, Ontario, Canada. [25]Division of Cardiology, Peter Munk Cardiac Centre, University Health Network, Toronto, Ontario, Canada. [26]Department of Computing, Imperial College London, London, UK. [27]Department of Brain Sciences, Imperial College London, London, UK. [28]Data Science Institute, Imperial College London, London, UK. [29]National Heart Centre, Singapore, Singapore. [30]Duke-National University of Singapore Medical School, Singapore, Singapore. [31]Department of Cardiology, Thorax Center, Cardiovascular Institute, Erasmus Medical Center, Rotterdam, the Netherlands. [32]Department of Genetics, University of Groningen, University Medical Center Groningen, Groningen, the Netherlands. [33]Department of Medicine, Cardiovascular Division, University of Virginia Health, Charlottesville, VA, USA. [34]Cardiovascular Division, Brigham and Women's Hospital, Boston, MA, USA. [35]Radcliffe Department of Medicine, University of Oxford, Division of Cardiovascular Medicine, NIHR Oxford Biomedical Research Centre, Oxford, UK. [36]UK Dementia Research Institute, Imperial College London, London, UK. [37]ECGen, Cardiogenetics Focus Group of EHRA, Biot, France. [38]Meyer Children's Hospital IRCCS, Florence, Italy. [39]Department of Medicine, University of Toronto, Toronto, Ontario, Canada. [40]Program in Medical and Population Genetics, Broad Institute of MIT and Harvard, Cambridge, MA, USA. [81]These authors contributed equally: Rafik Tadros, Sean L. Zheng, Anuj Goel, James S. Ware, Connie R. Bezzina, Hugh Watkins. *A list of authors and their affiliations appears at the end of the paper. ✉e-mail: rafik.tadros@umontreal.ca; j.ware@imperial.ac.uk; c.r.bezzina@amsterdamumc.nl; hugh.watkins@rdm.ox.ac.uk

## Hypergenes InterOmics Collaborators

**Daniele Cusi[41], Paolo Manunta[42], Lorena Citterio[42] & Nicola Glorioso[43]**

[41]Institute of Biomedical Technologies Milano National Research Council of Italy (CNR), Milan, Italy. [42]Genomics of Renal Diseases and Hypertension Unit, Istituto di Ricovero e Cura a Carattere Scientifico IRCCS San Raffaele Scientific Institute, Vita-Salute San Raffaele University, Milan, Italy. [43]Department of Clinical and Experimental Medicine, Hypertension and Related Diseases Centre, University of Sassari, Sassari, Italy.

## HCMR Investigators

**Theodore Abraham[44], Lisa Anderson[45], Florian Andre[46], Evan Appelbaum[47], Camillo Autore[48], Lauren Baldassarre[49], Colin Berry[50], Elena Biagini[51], William Bradlow[52], Chiara Bucciarelli-Ducci[53], Amedeo Chiribiri[23], Lubna Choudhury[54], Andrew Crean[39], Dana Dawson[55], Milind Desai[56], Patrice Desvigne-Nickens[57], John DiMarco[33], Eleanor Elstein[58], Andrew Flett[59], Matthias Friedrich[58], Eli Gelfand[47], Nancy Geller[57], Tjeerd Germans[60], Jeffrey Geske[61], Allison Hays[44], Stephen B. Heitner[62], Adam Helms[63], Carolyn Y. Ho[34], Daniel Jacoby[49], Dong-Yun Kim[57], Bette Kim[64], Han Kim[65], Paul Kolm[66], Christopher M. Kramer[33], Raymond Kwong[34], Eric Larose[67], Christopher Madias[68], Masliza Mahmod[7], Heiko Mahrholdt[69], Martin Maron[68], Ahmad Masri[62], Gerry McCann[70], Michelle Michels[14,31], Saidi Mohiddin[71], Francois-Pierre Mongeon[1], Sherif Nagueh[72], Stefan Neubauer[35], David Newby[73], Angus Nightingale[53], Iacopo Olivotto[38], Anjali Owens[74], Sven Plein[75], Sanjay K. Prasad[4,6], Betty Raman[7], Ornella Rimoldi[76], Michael Salerno[33], Jeanette Schulz-Menger[77], Sanjay Sharma[45], Mark Sherrid[78], Albert van Rossum[60], Hugh Watkins[7,81], Jonathan Weinsaft[79], William Weintraub[66], James White[80], Eric Williamson[61] & Anna Woo[39]**

[44]Johns Hopkins University, Baltimore, MD, USA. [45]St-George's University of London, London, UK. [46]University Hospital Heidelberg, Heidelberg, Germany. [47]Beth Israel Deaconess Medical Center, Boston, MA, USA. [48]Sapienza University of Rome, Rome, Italy. [49]Yale School of Medicine, New Haven, CT, USA. [50]University of Glasgow, Glasgow, Scotland. [51]University of Bologna, Bologna, Italy. [52]University Hospital Birmingham, Birmingham, UK. [53]University of Bristol, Bristol, UK. [54]Northwestern University, Chicago, IL, USA. [55]University of Aberdeen, Aberdeen, UK. [56]Cleveland Clinic, Cleveland, OH, USA. [57]National Heart Lung & Blood Institute, Bethesda, MD, USA. [58]McGill University, Montreal, Quebec, Canada. [59]University Hospital Southampton, Southampton, UK. [60]Amsterdam Cardiovascular Sciences, Vrije Universiteit Amsterdam, Amsterdam UMC, Amsterdam, the Netherlands. [61]Mayo Clinic, Rochester, MN, USA. [62]Oregon Health & Science University, Portland, OR, USA. [63]University of Michigan, Ann Arbor, MI, USA. [64]Icahn School of Medicine at Mount Sinai, New York, NY, USA. [65]Duke University, Durham, NC, USA. [66]MedStar Health Research Institute, Washington, DC, USA. [67]Quebec Heart and Lung Institute, Quebec City, Quebec, Canada. [68]Tufts Medical Center, Boston, MA, USA. [69]Robert-Bosch-Krankenhaus, Stuttgart, Germany. [70]University of Leicester, Leicester, UK. [71]Barts Heart Centre, London, UK. [72]Methodist DeBakey Heart and Vascular Center, Houston, TX, USA. [73]University of Edinburgh, Edinburgh, UK. [74]Perelman School of Medicine, University of Pennsylvania, Philadelphia, PA, USA. [75]University of Leeds, Leeds, UK. [76]San Raffaele Hospital, Milan, Italy. [77]Charité - Universitätsmedizin Berlin, Berlin, Germany. [78]New York University Langone Medical Center, New York, NY, USA. [79]Division of Cardiology, Weill Cornell Medicine, New York, NY, USA. [80]Libin Cardiovascular Institute, University of Calgary, Calgary, Alberta, Canada.

# Methods

## Ethics

The study was approved by the following ethics review boards: Research Ethics and New Technology Development Committee of the Montreal Heart Institute (2011/208), Medical Ethical Committee of Amsterdam University Medical Center (UMC; W20_226 no. 20.260), South Central–Hampshire B Research Ethics Committee (09/H0504/104), Hammersmith and Queen Charlotte's Research Ethics Committee (09/H0707/69) and the National Research Ethics Service (11/NW/0382, 13/EE/0325, 14/EE/1112, 14/SC/0190, 19/SC/0257, 21/NW/0157). The study of HCM patients from Amsterdam UMC was performed under a waiver—approved by the Medical Ethical Committee of Amsterdam UMC—allowing genotyping and genome-wide association study of people affected by cardiovascular disease. All other study participants provided informed consent.

## GWAS of HCM

The HCM GWAS included cases and controls from seven strata: the Hypertrophic Cardiomyopathy Registry (HCMR), a Canadian HCM cohort, a Netherlands HCM cohort, the Genomics England 100,000 Genome Project (GEL), the Royal Brompton HCM cohort, an Italian HCM cohort and the BioResource for Rare Disease (BRRD) project. Quality control (QC) and association analyses were performed per strata, followed by a meta-analysis. The seven strata are described in the Supplementary Note and in Supplementary Table 1. Cases consisted of unrelated patients diagnosed with HCM in presence of unexplained LV hypertrophy defined as a LV wall thickness (LVWT) > 15 mm, or >13 mm and either presence of family history of HCM or a pathogenic or likely pathogenic genetic variant causing HCM. HCM cases underwent gene panel sequencing as per clinical indications. Variants identified within eight core sarcomere genes (*MYBPC3*, *MYH7*, *TNNI3*, *TNNT2*, *MYL2*, *MYL3*, *ACTC1* and *TPM1*) were assessed centrally at the Oxford laboratory using the American College of Medical Genetics and Genomics (ACMG) guidelines[40]. HCM cases were dichotomized into sarcomere-positive and sarcomere-negative groups using a classification framework previously reported in Neubauer et al.[41]. In addition to the primary all-comer GWAS analyses including all cases with HCM (total of 5,900 cases and 68,359 controls), analyses stratified for sarcomere status in cases and randomly allocated controls were performed, including a total of 1,776 cases versus 29,414 controls in the $HCM_{SARC+}$ analysis and 3,860 cases versus 38,942 controls in the $HCM_{SARC-}$ analysis.

Meta-analyses for the all-comer HCM GWAS was performed on betas and standard errors using GWAMA[42]. We kept variants where meta-analysis came from two or more studies and also had a sample size >5,000. Genomic inflation was estimated from the median $\chi^2$ distribution and using HapMap3 European ancestry LD scores using LD Score Regression[5]. All variants were mapped to Genome Reference Consortium Human Build 37 (GRCh37) extrapolated using the 1000 Genome phase 3 genetic maps. A genome-wide significant locus was assigned where two variants had a meta-analysis $P < 5 \times 10^{-8}$ and were 0.5 cM distance apart. A similar approach was implemented for the $HCM_{SARC+}$ and $HCM_{SARC-}$ stratified analyses, which comprised five and seven strata, respectively (the GEL and BRRD strata did not include enough $HCM_{SARC+}$ cases). Variants were retained where meta-analysis came from two or more studies and had sample size >5,000 for $HCM_{SARC-}$ and >2,500 for $HCM_{SARC+}$ cases. The final dataset included 9,492,702 (all-comer), 7,614,734 ($HCM_{SARC+}$) and 9,226,079 ($HCM_{SARC-}$) variants after filtering. The results of the all-comer HCM GWAS meta-analysis and stratified analyses are presented in Table 1, Supplementary Figs. 1 and 2 and Supplementary Table 2.

A FDR 1% $P$ value cut-off was derived from the all-comer, $HCM_{SARC+}$ and $HCM_{SARC-}$ summary statistics using Simes method (Stata v.10.1), and the corresponding $P$ values were $8.5 \times 10^{-6}$, $1.6 \times 10^{-6}$ and $7.8 \times 10^{-6}$, respectively. Using the 1% FDR $P$-value thresholds, we then performed a stepwise model selection to identify 1% FDR independently associated variants using GCTA[4]. The analysis was performed chromosome wise using default window of 10 Mb, 0.9 collinearity and UKB reference panel containing 60,000 unrelated European ancestry participants. The results of this conditional analysis are presented in Supplementary Table 3.

## HCM heritability attributable to common variants

We estimated the heritability of HCM attributable to common genetic variation ($h^2_{SNP}$) in the all-comer HCM, as well as $HCM_{SARC+}$ and $HCM_{SARC-}$ using LDSC[5] and GREML[6]. For LDSC, HapMap3 SNPs were selected from the summary statistics corresponding to HCM, $HCM_{SARC+}$ and $HCM_{SARC-}$ meta-analyses. The $h^2_{SNP}$ was computed on the liability scale assuming a disease prevalence of 0.002 (ref. [43]). Since LDSC tends to underestimate $h^2_{SNP}$, we also estimated $h^2_{SNP}$ using GREML, as previously performed[2,3]. We first computed $h^2_{SNP}$ for HCM, $HCM_{SARC+}$ and $HCM_{SARC-}$ using GREML for each of the largest three strata (HCMR, the Canadian HCM cohort and the Netherlands HCM cohort), followed by fixed-effects and random-effects meta-analyses combining all three strata. To exclude the contribution of rare founder HCM causing variants, we excluded the *MYBPC3* locus for the Canadian and Netherlands strata and the *TNNT2* locus for the Canadian stratum[3]. The results of $h^2_{SNP}$ analyses are presented in Supplementary Table 4.

## GWAS of CMR imaging-derived LV traits

**UKB study population.** The UKB is an open-access population cohort resource that has recruited half a million participants in its initial recruitment phase, from 2006 to 2010. At the time of analysis, CMR imaging data were available from 39,559 participants in the imaging substudy. The UKB CMR acquisition protocol has been described previously[44]. In brief, images were acquired according to a basic cardiac imaging protocol using clinical 1.5 T wide bore scanners (MAGNETOM Aera, Syngo Platform VD13A, Siemens Healthcare) in three separate imaging centers. Extensive clinical and questionnaire data and genotypes are available for these participants. Clinical data were obtained at the time of the imaging visit. These included sex (31), age (21003), weight (21002), height (50), SBP (4080), DBP (4079), self-reported noncancer illness code (20002) and ICD-10 codes (41270). The mean age at the time of CMR was 63 ± 8 years (range 45–80) and 46% of participants were male. Cohort anthropometrics, demographics and comorbidities are reported in Supplementary Table 5. Exclusion criteria for the UKB imaging substudy included childhood disease, pregnancy and contraindications to magnetic resonance imaging (MRI) scanning. For the current analysis, we also excluded, by ICD-10 code and/or self-reported diagnoses, anyone with heart failure, cardiomyopathy, a previous myocardial infarction or structural heart disease. After imaging QC and exclusions for comorbidities or genotype QC, we had a maximum cohort size of 36,083 people. The UKB received National Research Ethics Approval (REC reference no. 11/NW/0382). The present study was conducted under terms of UKB access approval no. 18545.

**LV trait phenotyping.** Description of CMR image analysis is detailed in the Supplementary Note and in ref. [3]. We included ten LV phenotypes for GWAS analyses: end-diastolic volume (LVEDV), end-systolic volume (LVESV), LV ejection fraction (LVEF), mass (LVM), concentricity index (LV concentricity index (LVconc) = LVM/LVEDV), mean wall thickness (meanWT) and maximum wall thickness (maxWT) as well as global peak strain in radial (strain^rad), longitudinal (strain^long) and circumferential (strain^circ) directions. The means and s.d. values of all ten LV phenotypes, overall and stratified by sex, are shown in Supplementary Table 5.

**LV trait GWAS.** A description of genotyping, imputation and QC appears in the Supplementary Note. The GWAS model for LVEF, LVconc, meanWT, maxWT, strain^rad, strain^long and strain^circ included age, sex, mean arterial pressure (MAP), body surface area (BSA, derived from the

Mosteller formula) and the first eight genotypic principal components as covariates. LVEDV, LVESV and LVM were indexed to BSA for the analysis, as commonly performed in clinical practice. For indexed values (LV end-diastolic volume indexed for BSA (LVEDVi), LV end-systolic volume indexed for BSA (LVESVi) and LV mass indexed for BSA (LVMi), the GWAS model did not include BSA as a covariate, but all other covariates were the same as for nonindexed phenotypes. BOLT-LMM (v.2.3.2)[45] was used to construct mixed models for association with around 9.5 million directly genotyped and imputed SNPs. A high-quality set of directly genotyped model SNPs was selected to account for random effects in the genetic association analyses. These were selected by MAF (>0.001), and LD-pruned ($r^2 < 0.8$) to create an optimum SNP set size of around 500,000. The model was then applied to the >9.8 million imputed SNPs passing QC and filtering. Results of the LV traits GWAS are shown in Supplementary Table 6 and Supplementary Figs. 3–12.

**Locus definition and annotation.** Genomic loci associated with all LV traits were annotated jointly. Specifically, summary statistics were combined with a *P* value corresponding to the minimal *P* value (minP) across all ten summary statistics. The minP summary statistics was then used to define loci using FUMA v.1.4.2 (ref. 14) using a maximum lead SNP *P* value of $5 \times 10^{-8}$, maximum GWAS *P* value of 0.05 and $r^2$ threshold for independent significant SNPs of 0.05 (using the European 1000 Genomes Project dataset) and merging LD blocks within 250 kb. Loci were then mapped to genes using positional mapping (<10 kb), expression quantitative trait loci mapping using GTEx v.8 restricted to atrial appendage, left ventricle and skeletal muscle tissues, and chromatin interaction mapping using left and right ventricles. Genes mapped using FUMA were further prioritized by querying the Clinical Genomes Resource (ClinGen)[46] for genes linked to Mendelian heart disease with moderate, strong or definitive evidence, and using a recent review of overlapping GWAS and Mendelian cardiomyopathy genes[26]. In addition to FUMA locus-to-gene mapping, we also report closest gene and top gene mapped using OpenTargets[13]. Annotated LV trait loci are shown in Supplementary Table 6.

## Genetic correlations between HCM and LV traits
Pairwise genetic correlations for HCM and the ten LV traits were assessed using LDSC v.1.0.1 (ref. 9). The analysis was restricted to well-imputed nonambiguous HapMap3 SNPs, excluding SNPs with MAF < 0.01 and those with low sample size, using default parameters. We then assessed genetic correlations for each of the 55 pairs (HCM and ten LV traits) using precomputed LD scores from the European 1000 Genomes Project dataset. We did not constrain the single-trait and cross-trait LD score regression intercepts. The results of the genetic correlation analyses are shown in Fig. 2 and Supplementary Table 7.

## Multitrait analysis of GWAS
We performed multitrait analysis of GWAS summary statistics using MTAG (v.1.0.8)[7] to increase power for discovery of genetic loci associated with HCM. MTAG jointly analyzes several sets of GWAS summary statistics of genetically correlated traits to enhance statistical power. Due to high computation needs to calculate the maxFDR with MTAG, we limited the number of GWAS summary statistics to four (HCM plus three LV traits). The three LV traits to include were selected as follows. First, we performed hierarchical clustering of the ten LV traits using the absolute value of the pairwise genetic correlations, Euclidean distance and the complete method, predefining the number of clusters to three. This resulted in clustering of LV traits into an LV contractility cluster (LVEF, strain^rad, strain^long and strain^circ), an LV volume cluster (LVEDVi, LVESVi) and an LV mass cluster (LVMi, LVconc, meanWT, maxWT) (Fig. 2). We then selected the trait with the highest genetic correlation with HCM from each cluster (strain^circ, LVESVi and LVconc) to include in MTAG together with HCM. Only SNPs included in all meta-analyses (that is, HCM and LV traits) were used in MTAG. The coded/noncoded

alleles were aligned for all four studies before MTAG, and multi-allelic SNPs were removed. All summary statistics refer to the positive strand of GRCh37 and, as such, ambiguous/palindromic SNPs (having alleles A/T or C/G) were not excluded. Regression coefficients (beta) and their s.e. were used as inputs for MTAG. The maxFDR was calculated as suggested by the MTAG developers[7]. MaxFDR calculates the type I error in the analyzed dataset for the worst-case scenario. We estimated the gain in statistical power by the increment in the $N_{eff}$. The $N_{eff}$ for the HCM GWAS was calculated using the following formula[7,47]:

$$N_{eff(GWAS)} = \frac{4}{N_{cases}^{-1} + N_{controls}^{-1}}$$

The $N_{eff}$ for the HCM MTAG was computed by means of the fold-increase in mean $\chi^2$, using the following formula[7], implemented in MTAG, where the MTAG $N_{eff}$ corresponds to the approximate sample size needed to achieve the same mean $\chi^2$ value in a standard GWAS:

$$N_{eff(MTAG)} = N_{eff(GWAS)} \times \left(\chi^2_{MTAG,mean} - 1/\chi^2_{GWAS,mean} - 1\right)$$

To explore whether HCM effects estimates derived from MTAG are accurate, we compared the regression coefficients derived from MTAG with those derived from GWAS. This was performed for all variants included in MTAG and GWAS, and for a subset of variants reaching nominal significance ($P < 0.001$) in either GWAS and/or MTAG (Supplementary Fig. 13). The results of HCM MTAG are presented in Fig. 3, Supplementary Table 8 and Supplementary Fig. 14.

## Genome-wide annotation and gene set enrichment analyses
Genome-wide analyses following MTAG were performed using MAGMA v.1.08, as implemented in FUMA[14], including gene set and tissue expression analyses. We used Gene Ontology gene sets from the Molecular Signatures Database (MsigDB, v.6.2) for the gene set analysis and GTEx v.8 for the tissue specificity analysis. The results of MAGMA analyses are shown in Supplementary Table 9 (gene set analyses) and Supplementary Table 10 (tissue specificity analyses).

## Cardiac cell type heritability enrichment analysis
Gene programs derived from snRNA-seq were used to investigate heritability enrichment in cardiac cell types and states using the sc-linker framework[12]. This approach uses snRNA-seq data to generate gene programs that characterize individual cell types and states. These programs are then linked to genomic regions and the SNPs that regulate them by incorporating Roadmap Enhancer-Gene Linking[48,49] and Activity-by-Contact models[50,51]. Finally, the disease informativeness of resulting SNP annotations was tested using stratified LDSC (S-LDSC)[52] conditional on broad sets of annotations from the baseline-LD model[53,54]. Cell type and state-specific gene programs were generated from snRNA-seq data of ventricular tissue from 12 control participants, with cell type and state annotations made as part of a larger study of ~880,000 nuclei (samples from 61 DCM and 12 control participants[11]). Cell states that may not represent true biological states (for example, technical doublets) were excluded from analysis. Results of sc-linker cardiac cell type heritability enrichment analysis are shown in Supplementary Fig. 15.

## Locus-to-gene annotation
A genome-wide significant HCM MTAG locus was assigned where two variants had a MTAG $P < 5 \times 10^{-8}$ and were 0.5 cM distance apart, as performed for the HCM GWAS. Prioritization of potential causal genes in HCM MTAG loci was performed using OpenTargets V2G mapping[13] and FUMA[14]. The lead SNP at each independent locus was used as input for OpenTargets V2G using the release of 12 October 2022. Locus-to-gene mapping with FUMA v.1.3.7 was performed based on (1) position (within 100 kb), (2) expression quantitative trait loci associations in

disease-relevant tissues (GTEx v.8 left ventricle, atrial appendage and skeletal muscle) and (3) chromatin interactions in cardiac tissue (left ventricle and right ventricle, FDR < $10^{-6}$).

We further annotated genes mapped using OpenTargets and/or FUMA with their implication in Mendelian cardiomyopathy. Specifically, we queried the Clinical Genome Resource (ClinGen[28,46]) for genes associated with any cardiomyopathy phenotype with a level of evidence of moderate, strong or definitive and included genes with robust recent data supporting an association with Mendelian cardiomyopathy[26].

We also prioritized genes based on RNA expression data from bulk tissue RNA-seq data in the GTEx[38] v.8 dataset accessible at the GTEx Portal and snRNA-seq data from Chaffin et al.[15] accessible through the Broad Institute Single Cell Portal (https://singlecell.broadinstitute.org/single_cell). Using the GTEx v.8 data, we assessed specificity of LV expression by computing the ratio of median LV transcripts per million (TPM) to the median TPM in other tissues excluding atrial appendage and skeletal muscle and averaging tissue within types (for example, all arterial tissues, all brain tissues and so on). High and Mid LV expression specificity were empirically defined as >10-fold and >1.5-fold LV to other tissues median TPM ratios, respectively. Using snRNA-seq data from Chaffin et al.[15], we report the expression in the cardiomyocyte_1 cell type using scaled mean expression (relative to each gene's expression across all cell types) and percentage of cells expressing. High and Mid expression in cardiomyocytes were empirically defined as percentage expressing cells ≥80% and 40–80%, respectively. Prioritized genes were defined as genes mapped using both OpenTargets (top three genes) and FUMA, and had either (1) High LV-specific expression, (2) High cardiomyocyte expression or (3) both Mid LV-specific expression and Mid cardiomyocyte expression.

Lead variants in MTAG and GWAS loci were also annotated using the Ensembl Variant Effect Predictor (VEP) and lookup of lead variants and variants in LD ($r^2 > 0.5$) in other GWAS was performed using OpenTargets genetics.

Gene mapping and variant annotation data are shown in Supplementary Table 11 (VEP annotation), Supplementary Table 12 (OpenTargets genes), Supplementary Table 13 (FUMA genes) and Supplementary Table 15 (lookup in other GWAS). Prioritized genes are illustrated in Fig. 4a.

### Transcriptome-wide association study

We used MetaXcan to test the association between genetically predicted gene expression and HCM using summary results from MTAG analysis[24,55]. Biologically informed MASHR-based prediction models of gene expression for LV and AA tissue from GTEx v.8 (ref. 56) were analyzed individually with S-PrediXcan[55], and then analyzed together using S-MultiXcan[24]. GWAS MTAG summary statistics were harmonized and imputed to match GTEx v.8 reference variants present in the prediction model. To account for multiple testing, TWAS significance was adjusted for the total number of genes present in S-MultiXcan analysis (13,558 genes, $P = 3.7 \times 10^{-6}$). TWAS results are shown in Supplementary Table 14.

### Association of rare LoF variants in prioritized genes with HCM

We assessed the association of rare LoF variants in each of 26 prioritized genes (Fig. 4a) with HCM using burden analysis in three primary cohorts (BRRD, GEL and UKB) followed by fixed-effect meta-analysis. For BRRD, HCM cases were probands within the bio-resource project HCM. Controls were all remaining participants within the BRRD projects excluding those also recruited into the GEL and GEL2 projects (Genomics England pilot data). For GEL, HCM cases were probands referred into GEL with a primary clinical diagnosis of HCM. Controls were probands without any primary or secondary cardiovascular disease or myopathy. For UKB, HCM cases were identified from self-reported questionnaires at study recruitment, ICD-10 codes from clinical admission data and death registries, and CMR imaging for the

subset of the cohort who underwent cardiac MRI testing (LV maximum wall thickness >15 mm). All participants with aortic stenosis were excluded from UKB cases. Sequencing data were available for only *SVIL* in the Oxford Medical Genetics Laboratory (OMGL), where cases were clinically diagnosed with HCM and referred for diagnostic panel testing. The control group for the OMGL analysis consisted of 5,000 white British ancestry and unrelated control participants selected randomly from the UKB; these participants had normal LV volume and function and no clinical diagnosis of any cardiomyopathy. The remaining UKB samples were used as controls for UKB burden analysis. Genetic variants were identified using next-generation sequencing (whole-genome sequencing for BRRD, GEL, panel/exome sequencing for OMGL cases and UKB) and annotated using VEP and LOFTEE plugin[57]. LoF variants were defined as those with the following VEP terms: stop lost, stop gained, splice donor variant, splice acceptor variant and frameshift variant. Only variants with a MAF < $10^{-4}$ in the non-Finnish European (NFE) ancestral group of gnomAD v.2.1.1 (ref. 58) were selected. LoF variants present in the Matched Annotation from NCBI and EMBL-EBI (MANE)/canonical transcript or next best transcript were retained for the analysis. The proportion of cases and controls with LoF variants were compared using the Fisher Exact test for each of the BRRD, GEL and OMGL datasets. UKB LoF burden test was performed using their REGENIE workflow. We included high-quality sequenced variants where >90% samples had a sequencing depth >10, and tested genes with a minor allele count of ≥10. Firth correction was used to account for inflation resulting from case–control imbalance in the UKB. As a negative control, we also performed association testing of rare (MAF < $10^{-4}$) synonymous variants for each of the 26 prioritized genes using an identical methodology. Meta-analysis of burden test results was performed using the IVW method including studies with no zero counts and estimating standard error using sample counts[59]. The results of LoF and synonymous variant association with HCM are shown in Supplementary Table 16 and Fig. 4b. Further results for *SVIL* LoF variant analyses are shown in Fig. 4c, Supplementary Fig. 19 and Supplementary Table 19.

An exploratory exome-wide gene-based burden testing for LoF variants was also performed, using two MAF thresholds (<$10^{-4}$ and <$10^{-3}$). For UKB, this exploratory exome-wide analysis was performed as for the targeted analysis described above. For GEL and BRRD, this analysis was performed on the corresponding GEL and BRRD servers, using pre-annotated (VEP) files and filtering for LoF variants with gnomAD NFE AF < $10^{-3}$. For BRRD, variants were lifted to human genome build GRCh38 and LOFTEE was used to select high confidence LoF variants. For OMGL, only rare *SVIL* variants were available for analysis. Gene-based rare variant burden analyses followed by meta-analysis were performed as mentioned in the preceding paragraph. A sample size weighted meta-analysis was also performed, using the $N_{eff}$ formula shown above. The results of these exploratory exome-wide gene-based analyses are shown in Supplementary Tables 17 and 18 (full summary statistics) and Supplementary Figs. 16–18 (quantile–quantile and Manhattan plots).

### Locus colocalization in DCM and HCM

We explored colocalization of HCM and DCM loci using GWAS-PW[33]. The genome was split into 1,754 approximately independent regions and the all-comer HCM meta-analysis results were analyzed with those of a publicly available DCM GWAS[34] using a Bayesian approach. GWAS-PW fits each locus into one of the four models where model 1 is association in only the first trait, model 2 is association in only the second trait, model 3 when the two traits colocalize, and model 4 when the genetic signals are independent in the two traits. We considered a locus to show colocalization when either trait harbors a genetic signal with $P < 1 \times 10^{-5}$ and the GWAS-PW analysis demonstrates a posterior probability of association for model 3 (PPA3) >0.8. Results of GWAS-PW are presented in Supplementary Table 20.

## Two-sample MR

We assessed whether increased contractility and blood pressure are causally linked to increased risk of HCM globally and its obstructive (oHCM) and non-obstructive (nHCM) forms using two-sample MR. LV contractility and blood pressure parameters were used as exposure variables, and HCM, oHCM and nHCM as outcomes. Analyses were performed using the TwoSampleMR (MRbase) package[60] (v.0.5.6) in R (v.4.2.0). Four exposure variables corresponding to measures of LV contractility were used separately: LVEF as a volumetric marker of contractility, and global strain (strain$^{circ}$, strain$^{rad}$ and strain$^{long}$) as contractility markers based on myocardial tissue deformation. Instrument SNPs for contractility were selected based on the LV trait GWAS presented here using a $P$ value threshold of $<5 \times 10^{-8}$. Only independent SNPs (using $r^2 < 0.01$ in the European 1000 Genomes population) were included. Instrument SNPs for the blood pressure analysis were selected with a similar approach using a published blood pressure GWAS[39]. The outcome summary statistics were those of the single-trait HCM case–control meta-analysis (5,900 cases and 68,359 controls). We also performed a GWAS meta-analysis including data from HCMR and the Canadian HCM cohort (Supplementary Table 1) for nHCM (2,491 cases and 27,109 controls) and oHCM (964 cases and 27,163 controls) to use as outcomes. For these stratified analyses, oHCM was defined as HCM in presence of a LV outflow tract gradient ≥30 mmHg at rest or during Valsalva/exercise at any timepoint. All other HCM cases were considered nHCM. Loci reaching $P < 5 \times 10^{-8}$ in oHCM and nHCM are shown in Supplementary Table 2 and lookup of all-comer HCM MTAG loci in oHCM and nHCM are shown in Supplementary Table 8.

Insertions/deletions and palindromic SNPs with intermediate allele frequencies (MAF > 0.42) were excluded, and other SNPs in the same locus were included only if $P < 5 \times 10^{-8}$. An inverse variance weighted MR model was used as a primary analysis. We used three additional methods as sensitivity analyses: weighted median, weighted mode and MR Egger. Cochran's $Q$ statistics were calculated to investigate heterogeneity between SNP causal effects using IVW. Evidence of directional pleiotropy was also assessed using the MR Egger intercept. Mean $F$-statistics were calculated to assess the strength of the genetic instruments used. Leave-one-out analyses were also performed to ensure the SNP causal effects are not driven by a particular SNP. To further explore the impact of pleiotropy in the contractility/HCM MR analysis and to evaluate the consequence of excluding outlier SNPs, we used the MR pleiotropy residual sum and outlier (MR-PRESSO) analysis[61]. MR-PRESSO consists of three steps: testing for horizontal pleiotropy (global test), correcting for horizontal pleiotropy using outlier removal (outlier test) and evaluating differences in the causal estimate before and after outlier removal (distortion test). The summary results of MR analyses and sensitivity analyses are shown in Fig. 5 and Supplementary Table 21, with effect plots shown in Supplementary Fig. 20 (contractility) and Supplementary Fig. 22 (blood pressure), and leave-one-out analyses for the contractility MR in Supplementary Fig. 21. The MR effects are shown per unit change (percentage for contractility; mmHg for blood pressure) in Supplementary Table 21 and Supplementary Figs. 20–22, and per s.d. change in Fig. 5. OR per s.d. increase are calculated as $OR = e^{\beta_{MR} \times s.d.}$; s.d. values are reported in Supplementary Table 21 and correspond to those in the current UKB CMR dataset (for contractility) and those reported by Evangelou et al.[39] in the UKB (for blood pressure).

### Reporting summary

Further information on research design is available in the Nature Portfolio Reporting Summary linked to this article.

## Data availability

Data from the Genome Aggregation Database (gnomAD, v.2.1.1) are available at https://gnomad.broadinstitute.org. Data from the UKB can be requested from the UKB Access Management System (https://bbams.ndph.ox.ac.uk). Data from the GTEx consortium are available at the GTEx Portal (https://gtexportal.org). Published snRNA-seq data are available at the Broad Single Cell Portal (https://singlecell.broad-institute.org/) and at the Cellxgene tool website (https://cellxgene.cziscience.com/collections/e75342a8-0f3b-4ec5-8ee1-245a23e0f7cb/private). The Genome assembly GRCh37 can be accessed using https://www.ncbi.nlm.nih.gov/datasets/genome/GCF_000001405.13/. Individual-level data sharing is subject to restrictions imposed by patient consent and local ethics review boards. Full GWAS summary statistics of HCM, HCM$_{SARC-}$, HCM$_{SARC+}$, MTAG and ten LV traits are available on the GWAS catalog (accession IDs GCST90435254–GCST90435267) and can be accessed interactively at www.well.ox.ac.uk/hcm.

## Code availability

The analyses reported in this manuscript rely on previously published software, as detailed in Methods and in the Reporting Summary.

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

## Acknowledgements

This work was supported by funding from the British Heart Foundation (BHF, RG/18/9/33887, RE/18/4/34215, FS/15/81/31817); the National Heart, Lung and Blood Institute (NIH grant U01HL117006-01A1); the Wellcome Trust (201543/B/16/Z, 107469/Z/15/Z, 200990/A/16/Z); the Wellcome Trust core awards (090532/Z/09/Z, 203141/Z/16/Z); the National Institute for Health Research (NIHR) Oxford Biomedical Research Centre; the NIHR Imperial College Biomedical Research Centre; NIHR Royal Brompton Cardiovascular Biomedical Research Unit; Sir Jules Thorn Charitable Trust (21JTA); the Medical Research Council (MRC, UK); the Dutch Heart Foundation (CVON 2018-30 PREDICT2); the Horstingstuit Foundation; the Montreal Heart Institute Foundation; the Philippa and Marvin Carsley Cardiology Chair; the Fonds de la Recherche du Québec-Santé (254616, 265449); and the Canadian Institutes for Health Research (CIHR, 428321). For the purpose of open access, the authors have applied a CC BY public copyright licence to any Author Accepted Manuscript version arising from this submission. The views expressed in this work are those of the authors and not necessarily those of the funders. R.T. holds the Canada Research Chair in translational cardiovascular genetics. S.L.Z. received support from BHF Centre of Research Excellence Clinical Research Fellowship (RE/18/4/34215). C.F. received support from a BHF Clinical Research Training Fellowship (FS/15/81/31817). S.J.J. was supported by a Junior Clinical Scientist Fellowship (03-007-2022-0035) from the Dutch Heart Foundation and by an Amsterdam UMC Doctoral Fellowship. A.R.H. received support from the MRC Doctoral Training Partnership. X.X. is currently a postdoctoral scientist supported by MRC Laboratory of Medical Sciences. K.A.M. is supported by a BHF Immediate Fellowship (FS/IPBSRF/22/27059). R.W. received support from an Amsterdam Cardiovascular Sciences fellowship. A.d.M. is supported by the Fetal Medicine Foundation (495237). M.T. receives support from Monat Foundation. Y.M.P. receives support from the Dutch Heart Foundation (CVON PRIME). B.P.H. is supported by the BHF Intermediate Fellowship (FS/ICRF/21/26019). D.P.O. is supported by the MRC (MC_UP_1605/13) and BHF (RG/19/6/34387, CH/P/23/80008). R.A.d.B. is supported by the Dutch Heart Foundation (2020B005), by the Leducq Foundation (Cure-PLaN) and by the European Research Council (ERC CoG 818715). I.C. receives support from the Dutch Heart Foundation (CVON 2015-12 eDETECT). P.M.M. received funding from the Edmond J. Safra Foundation and Lily Safra and an NIHR Senior Investigator Award, the UK Dementia Research Institute, which receives its funding from UK DRI Ltd., funded by the MRC, Alzheimer's Society, Alzheimer's Research UK and the Imperial College British Heart Foundation Centre of Excellence. J.-C.T. holds the Canada Research Chair in personalized medicine and the University of Montreal endowed research chair in atherosclerosis, and is also the principal investigator of the Montreal Heart Institute André and France Desmarais hospital cohort funded by the Montreal Heart Institute Foundation. A.G. has received support from the BHF, European Commission (LSHM-CT- 2007-037273, HEALTH-F2-2013-601456) and TriPartite Immunometabolism Consortium (TriC)-Novo Nordisk Foundation (NNF15CC0018486), BHF-DZHK (SP/19/2/344612). C.R.B. received support from EJP-RD (LQTS-NEXT, ZonMW project 40-46300-98-19009) and the Leducq Foundation (project 17CVD02). H.W. is member of the Oxford BHF Centre for Research Excellence (RE/13/1/30181). H.W. and J.S.W. are supported by CureHeart, the British Heart Foundation's Big Beat Challenge award (BBC/F/21/220106). This research has been conducted in part using the UK Biobank Resource under application nos. 18545 and 47602. This research was made possible through access to the data and findings generated by the 100,000 Genomes Project. The 100,000 Genomes Project is managed by Genomics England Limited (a wholly owned company of the Department of Health and Social Care) and funded by the National Institute for Health Research and NHS England. The Wellcome Trust, Cancer Research UK and the Medical Research Council have also funded research infrastructure. The 100,000 Genomes Project uses data provided by patients and collected by the National Health Service as part of their care and support. The Genotype-Tissue Expression (GTEx) Project was supported by the Common Fund of the Office of the Director of the National Institutes of Health, and by NCI, NHGRI, NHLBI, NIDA, NIMH and NINDS. The data used for the analyses described in this manuscript were obtained from the GTEx Portal. Genotyping of the Florence Cohort was supported by the Institute of Psychiatry Psychology and Neuroscience (IoPPN) Genomics and Biomarker Core Facility within King's College London, who gratefully acknowledge capital equipment funding from the Maudsley Charity (980) and Guy's and St Thomas's Charity (STR130505).

## Author contributions

R.T., S.L.Z., C.G., P.J., C.F., A.G., J.S.W., C.R.B. and H.W. conceived or designed elements of the study. R.T., S.L.Z., C.G., P.J., C.F., D.M.W., S.J.J., K.L.T., A.R.H., E.O., X.X., P.I.T., R.J.B., K.A.M., F.M., B.B., E.P., M.L., M.N., A.V., A.M.C.V., R.W., A.S.A., M.A.vS., N.M.R., L.J.S., E.S., C.L., A.d.M., J.D.R., M.T.-G., G.G., J.C.-T., P.L.L., P.G., M.T., A.A.G.T., Y.M.P., H.R., A.P., W.B., J.B., B.P.H., S.K.P., P.J.R.B., D.P.O., S.A.C., R.A.d.B., I.C., M.M., C.M.K., C.Y.H., S.N., P.M.M., A.A.M.W., J.-C.T., I.O., A.A., A.G., J.S.W., C.R.B. and H.W. acquired, analyzed or interpreted data. R.T., S.L.Z., C.G., P.J., C.F., A.G., J.S.W., C.R.B. and H.W. drafted the manuscript. All authors critically revised the manuscript for important intellectual content and approved the final version.

## Competing interests

R.T. has received research support and consultancy fees from Bristol Myers Squibb. A.R.H. is a current employee and stockholder of AstraZeneca. D.P.O. has received grants and consultancy fees from Bayer. R.d.B. has received research grants and/or fees from AstraZeneca, Abbott, Boehringer Ingelheim, Cardior Pharmaceuticals GmbH, Ionis Pharmaceuticals, Inc., Novo Nordisk and Roche, and also has speaker engagements with Abbott, AstraZeneca, Bayer, Bristol Myers Squibb, Novartis and Roche. P.G. receives research funds from Abbott Cardiovascular and Medtronics. M.M. has received research support or consultancy fees from Bristol Myers Squibb, Cytokinetics, Pfizer, Sanofi Genzyme, Biomarin and Alnylam. C.M.K. received research grants from Cytokinetics and Bristol Myers Squibb. P.M.M. has received consultancy fees from Roche, Biogen, Nodthera and Sangamo Pharmaceuticals and has received research or educational funds from Biogen, Novartis, Merck and Bristol Myers Squibb. J.-C.T. has received research grants from Amarin, AstraZeneca, Ceapro, DalCor, Esperion, Ionis, Novartis, Pfizer and RegenXBio; honoraria from AstraZeneca, DalCor, HLS Therapeutics, Pendopharm and Pfizer; holds minor equity interest in DalCor; and is an author of a patent on pharmacogenomics-guided CETP inhibition. J.S.W. has received research support or consultancy fees from Myokardia, Bristol Myers Squibb, Pfizer and Foresite Labs. C.R.B. has consulted for Illumina. H.W. has consulted for Cytokinetics, BridgeBio and BioMarin. The remaining authors declare no competing interests.

## Additional information

**Correspondence and requests for materials** should be addressed to Rafik Tadros, James S. Ware, Connie R. Bezzina or Hugh Watkins.

# Reporting Summary

## Statistics

For all statistical analyses, confirm that the following items are present in the figure legend, table legend, main text, or Methods section.

| n/a | Confirmed | |
|---|---|---|
| ☐ | ☒ | The exact sample size (*n*) for each experimental group/condition, given as a discrete number and unit of measurement |
| ☐ | ☒ | A statement on whether measurements were taken from distinct samples or whether the same sample was measured repeatedly |
| ☐ | ☒ | The statistical test(s) used AND whether they are one- or two-sided<br>*Only common tests should be described solely by name; describe more complex techniques in the Methods section.* |
| ☐ | ☒ | A description of all covariates tested |
| ☐ | ☒ | A description of any assumptions or corrections, such as tests of normality and adjustment for multiple comparisons |
| ☐ | ☒ | A full description of the statistical parameters including central tendency (e.g. means) or other basic estimates (e.g. regression coefficient) AND variation (e.g. standard deviation) or associated estimates of uncertainty (e.g. confidence intervals) |
| ☐ | ☒ | For null hypothesis testing, the test statistic (e.g. *F*, *t*, *r*) with confidence intervals, effect sizes, degrees of freedom and *P* value noted<br>*Give P values as exact values whenever suitable.* |
| ☒ | ☐ | For Bayesian analysis, information on the choice of priors and Markov chain Monte Carlo settings |
| ☒ | ☐ | For hierarchical and complex designs, identification of the appropriate level for tests and full reporting of outcomes |
| ☐ | ☒ | Estimates of effect sizes (e.g. Cohen's *d*, Pearson's *r*), indicating how they were calculated |

*Our web collection on statistics for biologists contains articles on many of the points above.*

## Software and code

Policy information about availability of computer code

| Data collection | No software used for data collection. |
|---|---|
| Data analysis | The following publicly available software were used in data analysis:<br>PLINK 1.9 (https://www.cog-genomics.org/plink)<br>SNPTEST 2.5 (https://www.well.ox.ac.uk/~gav/snptest)<br>SAIGE v0.29.4.2 (https://github.com/weizhouUMICH/SAIGE)<br>SHAPEIT v2.r790 (https://mathgen.stats.ox.ac.uk/genetics_software/shapeit/shapeit.html)<br>IMPUTE2 v2.3.2 (https://mathgen.stats.ox.ac.uk/impute/impute_v2.html)<br>Michigan Imputation Server v1.2.4 (https://imputationserver.sph.umich.edu)<br>GWAMA (https://genomics.ut.ee/en/tools)<br>STATA 10.1 (https://www.stata.com)<br>GCTA (https://yanglab.westlake.edu.cn/software/gcta)<br>LDSC v.1.0.1 (https://github.com/bulik/ldsc)<br>GWAS-PW (https://github.com/joepickrell/gwas-pw)<br>MIRTK toolkit (http://mirtk.github.io)<br>BOLT-LMM v2.3.2 (https://alkesgroup.broadinstitute.org/BOLT-LMM)<br>FUMA 1.5.1 (https://fuma.ctglab.nl)<br>MTAG v.1.0.8 (https://github.com/JonJala/mtag)<br>sc-linker (https://github.com/karthikj89/scgenetics)<br>OpenTargets, release of October 12th, 2022 (https://www.opentargets.org)<br>REGENIE (https://rgcgithub.github.io/regenie/) |

MetaXcan (https://github.com/hakyimlab/MetaXcan)
PrediXcan (https://github.com/hakyimlab/PrediXcan)
MR-base v.0.5.6 (https://www.mrbase.org)
R project v.4.2.0 (https://www.r-project.org)

For manuscripts utilizing custom algorithms or software that are central to the research but not yet described in published literature, software must be made available to editors and reviewers. We strongly encourage code deposition in a community repository (e.g. GitHub). See the Nature Portfolio guidelines for submitting code & software for further information.

## Data

Policy information about availability of data

All manuscripts must include a data availability statement. This statement should provide the following information, where applicable:
- Accession codes, unique identifiers, or web links for publicly available datasets
- A description of any restrictions on data availability
- For clinical datasets or third party data, please ensure that the statement adheres to our policy

Data from the Genome Aggregation Database (gnomAD, v.2.1.1) are available at https://gnomad.broadinstitute.org. Data from the UKB can be requested from the UKB Access Management System (https://bbams.ndph.ox.ac.uk). Data from the GTEx consortium are available at the GTEx portal (https://gtexportal.org). Published snRNA-seq data are available at the Broad Single Cell Portal (https://singlecell.broadinstitute.org/) and at the Cellxgene tool website (https://cellxgene.cziscience.com/collections/e75342a8-0f3b-4ec5-8ee1-245a23e0f7cb/private). The Genome assembly GRCh37 can be accessed using https://www.ncbi.nlm.nih.gov/datasets/genome/GCF_000001405.13/. Individual level data sharing is subject to restrictions imposed by patient consent and local ethics review boards. Full GWAS summary statistics of HCM, HCMSARC-, HCMSARC+, MTAG, and 10 LV traits are available on the GWAS catalog (accession IDs GCST90435254 to GCST90435267) and can be accessed interactively at www.well.ox.ac.uk/hcm.

## Human research participants

Policy information about studies involving human research participants and Sex and Gender in Research.

| | |
|---|---|
| Reporting on sex and gender | Biological sex is reported for all included study cohorts. Females constitute 31% of cases and 36% of controls in the HCM case-control GWAS, and 54% of UK Biobank participants with available cardiac MRI included in the left ventricular GWAS. Sex was included as a covariate in regression analyses. No sex-stratified analyses were performed. |
| Population characteristics | HCM GWAS cohort: mean age ranging from 51 to 67 across cohorts, median maximal left ventricular wall thickness 19mm, 32% carrying a presumably causal rare sarcomeric gene variant, 28% with obstructive physiology. Details are presented in Supplementary Table 1.<br><br>UK Biobank cardiac magnetic resonance cohort: mean age 63, mean body mass index 26, 25% with a diagnosis of hypertension. Details are presented in Supplementary Table 7. |
| Recruitment | Cases clinically diagnosed with HCM were enrolled from local, national or international studies. Controls without known HCM were included from population-matched cohorts. Details appear in the Supplementary Note. |
| Ethics oversight | All components of the study were approved by ethics review boards at corresponding institutions:<br>- Medisch Ethische Toetsingscommissie (METC) of the Amsterdam University Medical<br>- Hammersmith & Queen Charlotte's Research Ethics Committee<br>- South Central - Hampshire B Research Ethics Committee<br>- Research Ethics and New Technology Development Committee of the Montreal Heart Institute<br>- South Central - Oxford A Research Ethics Committee<br>- East of England - Cambridge South Research Ethics Committee |

Note that full information on the approval of the study protocol must also be provided in the manuscript.

# Field-specific reporting

Please select the one below that is the best fit for your research. If you are not sure, read the appropriate sections before making your selection.

☒ Life sciences　　　☐ Behavioural & social sciences　　　☐ Ecological, evolutionary & environmental sciences

For a reference copy of the document with all sections, see nature.com/documents/nr-reporting-summary-flat.pdf

# Life sciences study design

All studies must disclose on these points even when the disclosure is negative.

| | |
|---|---|
| Sample size | For hypertrophic cardiomyopathy (HCM), we analyzed the largest sample size available, including 5,900 HCM cases and 68,359 controls. For left ventricular (LV) traits, we analyzed the largest sample size with available cardiac magnetic resonance imaging studies from the UK Biobank at the time of analysis, consisting of 36,083 individuals. For association of rare loss of function variants with HCM, we included 1,845 HCM |

cases and 37,481 controls. No method was applied to predetermine sample size. Such sample size allowed the identification of significant associations for ALPK3 and SVIL but may have been underpowered for genes with smaller effect sizes.

| Data exclusions | Pre-established quality control processes were applied during inclusion (clinical and imaging data) and during analysis (genotypic data). |
| --- | --- |
| Replication | No replication of GWAS signals was attempted in the absence of a sufficiently powered independent cohort and considering the conventional significance threshold for GWAS set to P≤5e-8 in the meta-analysis. |
| Randomization | The GWAS used a case-control study design. In the process of selecting appropriate controls, the R package dplyr::sample_n was used to pseudo-randomly select controls. |
| Blinding | Blinding was not possible in the analysis plan. |

# Reporting for specific materials, systems and methods

We require information from authors about some types of materials, experimental systems and methods used in many studies. Here, indicate whether each material, system or method listed is relevant to your study. If you are not sure if a list item applies to your research, read the appropriate section before selecting a response.

### Materials & experimental systems

| n/a | Involved in the study |
| --- | --- |
| ☒ | Antibodies |
| ☒ | Eukaryotic cell lines |
| ☒ | Palaeontology and archaeology |
| ☒ | Animals and other organisms |
| ☒ | Clinical data |
| ☒ | Dual use research of concern |

### Methods

| n/a | Involved in the study |
| --- | --- |
| ☒ | ChIP-seq |
| ☒ | Flow cytometry |
| ☒ | MRI-based neuroimaging |

