## [Peer Review File · Nature Genetics]

Large scale genome-wide association analyses identify novel genetic loci and mechanisms in hypertrophic cardiomyopathy

Corresponding Author: Dr Rafik Tadros

Version 0:

Decision Letter:

6th April 2023

Dear Rafik,

Your Letter entitled "Large scale genome-wide association analyses identify novel genetic loci and mechanisms in hypertrophic cardiomyopathy" has been seen by three referees. You will see from their comments below that, while they find your work of potential interest, they have raised substantial concerns that must be addressed. In light of these comments, we cannot accept the manuscript for publication at this time, but we would be interested in considering a suitably revised version that addresses the referees' concerns.

We hope you will find the referees' comments useful as you decide how to proceed. If you wish to submit a substantially revised manuscript, please bear in mind that we will be reluctant to approach the referees again in the absence of major revisions.

To guide the scope of the revisions, the editors discuss the referee reports in detail within the team, including with the chief editor, with a view to identifying key priorities that should be addressed in revision, and sometimes overruling referee requests that are deemed beyond the scope of the current study. In this case, we ask that you address all technical queries related to the association analyses, particularly with respect to assessing heterogeneity across the study cohorts and interpreting the MTAG analyses, and that you extend the burden analyses to encompass a larger set of genes, if feasible. We hope you will find this prioritized set of referee points to be useful when revising your study. Please do not hesitate to get in touch if you would like to discuss these issues further.

If you choose to revise your manuscript taking into account all reviewer and editor comments, please highlight all changes in the manuscript text file. At this stage, we will need you to upload a copy of the manuscript in MS Word .docx or similar editable format.

*2) If you have not done so already, please begin to revise your manuscript so that it conforms to our Letter format instructions, available here. Refer also to any guidelines provided in this letter.

*3) Include a revised version of any required Reporting Summary: <https://www.nature.com/documents/nr-reporting-summary.pdf>

Link Redacted

If you wish to submit a suitably revised manuscript, we hope to receive it within 3-6 months. If you cannot send it within this time, please let us know. We will be happy to consider your revision so long as nothing similar has been accepted for publication at Nature Genetics or published elsewhere. Should your manuscript be substantially delayed without notifying us in advance and your article is eventually published, the received date would be that of the revised, not the original, version.

Nature Genetics is committed to improving transparency in authorship. As part of our efforts in this direction, we are now requesting that all authors identified as 'corresponding author' on published papers create and link their Open Researcher and Contributor Identifier (ORCID) with their account on the Manuscript Tracking System (MTS), prior to acceptance. ORCID helps the scientific community achieve unambiguous attribution of all scholarly contributions. You can create and link your ORCID from the home page of the MTS by clicking on 'Modify my Springer Nature account'. For more information, please visit www.springernature.com/orcid.

Thank you for the opportunity to review your work.

Sincerely,
Kyle

Kyle Vogan, PhD
Senior Editor
Nature Genetics
<https://orcid.org/0000-0001-9565-9665>

Referee expertise:

Referee #1: Genetics, cardiovascular diseases, cardiac MRI

Referee #2: Genetics, cardiovascular diseases, statistical methods

Referee #3: Genetics, cardiovascular diseases, cardiomyopathies

Reviewers' Comments:

Reviewer #1:
Remarks to the Author:

Large scale GWAS identify novel genetic loci and mechanisms in hypertrophic cardiomyopathy.

Tadros et al

Summary of paper

This paper describes the results from a GWAS of HCM (5000 HCM cases, 68,000 controls) and MTAG analysis including over 5000 HCM cases, 68,000 controls and c. 36,000 individuals with cardiac MRI data. The authors report 70 loci (50 novel) for HCM, 62 (32 novel) for LV traits measured using cardiac MRI. One novel gene is highlighted and discussed – supervillin. MR analysis exploring hypertension and LV contractility supports a causative role in both obstructive and non-obstructive forms of HCM.

Comments to authors

Increasing sample size from prior work by the co-authors with new cohorts has yielded many new loci from both GWAS of all HCM patients, sarcomere positive and negative subsets, and new additional loci using MTAG analysis.

There are several suggestions and points of clarification I would like to make:

1. GWAS of HCM – selection and information on the cases and controls – how was matching done and analysis? ST1 does not provide information on the controls? Can this information be included and ancestry matching, what are the demographics of the controls?
2. The MTAG method has been used to leverage additional loci associated with HCM. This is a method used in prior work by some of the authors for identifying HCM loci. This method provides loci which are important to validate. I wondered as this was a method used in prior work – how many of the loci discovered from the MTAG analysis in their prior study were genome-wide significant in this new meta analysis of HCM? The results from this would provide validation of this method for leveraging loci from this method for HCM and thus confidence in this new number of loci now being reported. Can these results be reported and discussed?
3. How reliable are MTAG estimates of effect sizes?
4. I was intrigued with the discovery and discussion of supervillin – indeed it is an interesting candidate but it was not genome-wide significant for all HCM, noting it was for MTAG (add this information to Figure 1). Follow up analyses provides support for this as a candidate gene for HCM and nicely demonstrates leveraging related traits. As there is no validation for other loci – is this an approach which you can use for potential validation of other interesting candidate genes from this work? This would add value to the manuscript as readers would be interested in knowing which genes may have rare variants and the data looks to be available if done for supervillin?
5. The all sample GWAS for HCM provided 50 novel loci – the authors do not provide a detailed follow up discussion of all potential candidate genes at these loci – can any further work be done and reported on? I note a lot of emphasis on supervillin from the pairwise GWAS as an interesting gene but not so much discussion on other genes from this work?
6. I was interested in the results from the GWAS of sarcomere (-) patients – 24 genome-wide significant loci are listed – opposed to the 70 in the full cohort. What is the overlap of loci are the same loci picked up – can you perform pathway analyses – do the results differ to results with all HCM patients?
7. From your work what is new beyond the prior GWAS on the biology of HCM?
8. Are potential candidate genes at loci also found for other cardiovascular traits? Is there pleiotropy? This is not explored or discussed?
9. You report new loci from the study of cardiac MRI traits – these results are not discussed or put in context with others work on genetic analysis of cardiac MRI traits? Can this be commented upon?
10. With MTAG leveraging loci for HCM – would it be fair to say the loci discovered from this analysis would also be loci for the cardiac MRI trait used as well – so the loci represent a dual phenotype, or am I misunderstanding this method? To rephrase my question – from the MTAG analysis reported previously by some of the co-authors, are loci also significant now in a GWAS of the cardiac MRI trait which was used in the MTAG?
11. You have performed slightly different downstream analysis from your different GWAS findings, but can your approach be commented on in the text – why was one pathway not followed?
12. The MR work was interesting but I was a little uncertain of how this fit in with this paper. It seemed like an analysis which was tagged on and perhaps be the topic of a different paper?
13. Are you able to perform genetic analysis of the obstructive and non-obstructive HCM into the paper if you then shift focus to this phenotype?

Reviewer #2:

Remarks to the Author:

The work by Tadros et al. is a large meta-analysis of hypertrophic cardiomyopathy (HCM) where the authors utilize GWASs from 7 cohorts (5,900 HCM cases) and perform GWASs of cardiac magnetic resonance (CMR) traits (N~36,000). They claim identification of 50 novel HCM loci and 32 novel loci associated with relevant left ventricular (LV) structural or functional traits. The authors claim SVIL as a novel HCM gene and explore HCM association with rare truncating variants in SVIL. Mendelian randomization analyses were conducted to explore the causal role of increased LV contractility in both obstructive (oHCM) and non-obstructive forms of HCM (nHCM). The work is well written and of interest, especially if the authors extend the Burden loss-of-function variant analysis beyond SVIL and perform it on all genes. This manuscript is somewhat a follow up of a paper published in 2021, a HCM GWAS utilizing ~ 1,700 HCM cases and LV traits (N=19,999) where they explored causal effects between increased LV contractility with HCM (not split into oHCM and nHCM).

Comments:

- 1) The authors use a genome-wide significance threshold of 5×10^{-8} . This threshold is somewhat outdated, and I would suggest using a Bonferroni threshold where the number of variants tested is accounted for, i.e. $0.05/9,492,702$. The authors do not have a replication set and they do not show if the associations are consistent across the cohorts they are using. I understand that replication cohorts are difficult to come by, but then it is more important to explore heterogeneity between the cohorts and be conservative when selecting the GWS threshold.
- 2) The authors perform a co-localization analysis searching for variants that associate with HCM and DCM with opposing genomic effects. They find an intergenic variant that associates suggestively (not genome-wide significant for HCM nor DCM) with HCM and DCM with opposing effects. In the analysis they require (somewhat arbitrarily) that the variants have a $P < 1 \times 10^{-5}$ for both phenotypes and have a posterior probability > 0.8 of colocalization. The authors acknowledge that this requires further evidence to support implication in HCM. This common variant association would then ideally be replicated in another HCM cohort. Somewhat surprisingly, the authors explore the closest gene (SVIL) in a burden analysis but this isn't a strict replication of the suggestive association. Did the common variant close to SVIL show consistent HCM association across cohorts?
- 3) The loss of function burden analysis suggests that SVIL loss-of-function variants cause HCM. The associations would survive a multiple testing adjustment where the number of genes in the genome would be accounted for. I would like to see this analysis not restricted to SVIL. I am sure that other genes in this analysis would be of interest and this would also help putting the SVIL association in perspective. Furthermore, the authors could provide QQ-plots etc. for QC analysis. They need to confirm that there is no inflation in test statistic etc. in the burden analysis.
- 4) Did the HCM cases that carried SVIL LOF variants also carry other pathogenic variants?
- 5) The authors utilize data from the UKB in the manuscript but not in the burden analysis. Several individuals carry SVIL loss-of-function variants in the UKB and HCM diagnostic ICD codes and imaging data are available. Is there any reason not to explore that?
- 6) Why use only variants with a $MAF < 10^{-4}$ in the non-Finnish European ancestral group of gnomAD v2.1.1? Are there other predicted loss-of-function variants found in the data that are more common?
- 7) Is there any available data to explore if the SVIL LOF heterozygotes have myofibrillar myopathy as described in OMIM for homozygotes?
- 8) The authors say that they "performed rare variant burden analysis including 1,845 clinically diagnosed unrelated HCM cases". Then they state "In one family, the SVIL LoF variant (p.(Gln255*)) was carried by two cousins with HCM". This is confusing and needs to be clarified.
- 9) The authors of the MTAG paper suggest using their method when the genetic correlation between traits is > 0.7 and state that "Replication is the best way to assess the credibility of individual-SNP associations." For HCM the traits have correlation < 0.7 and there is no replication cohort available. The authors do perform an estimation of the upper bound for the FDR, but this analysis holds under certain assumptions. Do these assumptions hold?
- 10) For the MTAG analysis, I would again suggest using a more conservative GWS threshold and show the HCM associations across all cohorts. Could the FinnGen dataset or UKB be useful?
- 11) The Mendelian randomization analysis is interesting and seems to be well conducted. There are however variants deviating from the regression line, e.g. variants that associate with LVEF but have no effect on oHCM or nHCM. Doesn't this question the claim of a causal relationship? Is there a reason to think that those instruments that deviate are invalid?
- 12) Doesn't the possibility of a bi-directional relationship between HCM and LV contractility need to be explored (also just HCM affecting LV contractility rather than vice versa)?
- 13) Why do the authors restrict the analysis to variants with $MAF > 1\%$?
- 14) There are likely some issues with the conditional analysis being performed, e.g. variants that are on chromosome 11 and are significant in the analysis are likely correlated and I suspect that they may not be associating if the authors would use genotypes rather than summary level data (COJO) to perform the conditional analysis.
- 15) The authors state that a "A locus on chromosome 11 which includes MYBPC3, a well-established disease gene, is associated with HCM and HCMSARC+, but not HCMSARC-, implying that this association is tagging known founder pathogenic variants in MYBPC3". Although this may be the case, I cannot see why these associations would imply that there is founder variant (A genetic alteration observed with high frequency in a group that is or was geographically or culturally isolated)? Do all variants that associate with HCMSARC+ and not HCMSARC- tag a founder variant?
- 16) I suggest annotating the variants using Variant effect predictor (VEP). I would e.g. like to know if any of the variants in main Table 1 are missense or loss-of-function.

Reviewer #3:

Remarks to the Author:

Summary of the key results

This is a large genome-wide association study (GWAS) and multi-trait analysis (MTAG) comprising a number of HCM cohorts (5,900 cases, 68,359 controls) and 36,083 UK Biobank (UKB) participants with cardiac magnetic resonance (CMR) imaging. The authors identify a total of 70 loci (50 novel) traits associated with HCM, and 62 loci (32 novel) associated with left ventricular (LV) structural or functional traits. Prominence is given to, SVIL, which encodes the actin-binding protein supervillin, making the suggestion that rare truncating SVIL variants cause HCM.

Originality and significance

There are already a number of GWAS studies in this area, including one by some of the same authors:

- Harper AR, Goel A, Grace C, Thomson KL, Petersen SE, Xu X, et al. Common genetic variants and modifiable risk factors underpin hypertrophic cardiomyopathy susceptibility and expressivity. *Nat Genet.* 2021;53(2):135-42.
- Tadros R, Francis C, Xu X, Vermeer AMC, Harper AR, Huurman R, et al. Shared genetic pathways contribute to risk of hypertrophic and dilated cardiomyopathies with opposite directions of effect. *Nat Genet.* 2021;53(2):128-34.
- Biddinger KJ, Jurgens SJ, Maamari D, Gaziano L, Choi SH, Morrill VN, et al. Rare and Common Genetic Variation Underlying the Risk of Hypertrophic Cardiomyopathy in a National Biobank. *JAMA Cardiol.* 2022;7(7):715-22.

The previous paper by Tadros has a similar design and conclusion. The only major difference is the size of the analysed cohort.

Data & methodology

The meta-analysis for "single-trait" is robust with some interesting hits. However, the paper focuses very heavily on the "new gene" SVIL - based on data solely from the GWAS. The same locus did not reach significance with the single trait analysis and a case vs control burden test for rare variants.

Supportive data on the significance of SVIL are few with little if any functional data (the only supporting information given is "cardiac abnormalities" in zebrafish).

Importantly, the authors mention a family where two cousins carry the variant, which is insufficient to support segregation. More data are required on this family and other patients in whom a Mendelian trait is genuinely suspected. As it stands, there is insufficient information to judge the veracity of the claim that SVIL is a new disease gene.

As there is not much biological evidence for SVIL, it is possible that this is a false discovery due to the fact that SVIL is a locus for the imaging trait jointly under study (which is possible as SVIL was picked up as a candidate gene for RV traits in another CMR GWAS (<https://www.nature.com/articles/s41588-022-01083-2>)).

The HCM cohorts enrolled in this study are very diverse and are likely to be subject to considerable ascertainment bias. The data on cohort characteristics are very sparse and under "HCM physiology available" absent. The data should be expanded to include a full description including missingness of the analysed traits. A sensitivity analysis to examine the effect of cohort/centre would be helpful. Even in the few data provided, there is a curious anomaly of fairly advanced age, which is unusual for HCM.

Appropriate use of statistics and treatment of uncertainties

As in a previous study by the same authors, the concept of the multi-trait analysis is a little difficult to understand. It is used to boost the power of the GWAS by doubling the number of loci through a combination of HCM-related loci with those that associate with functional and structural traits such as strain, ventricular volume and mass. While there may be overlap, I am not sure that we can say the latter are genuine HCM loci.

MTAG makes sense if the traits are closely correlated. For example a genotypic correlation > 0.7 . It is a little surprising to see the use of MTAG for a mix of all imaging traits plus HCM/DCM as it is likely the genotypic correlation will be weaker among these traits and type 1 error will be substantially inflated.

With regard to the ancestral composition of the case and control sets, the HCM cases were enrolled in a number of countries but the majority of the controls are from the UK. A sensitivity analysis restricted to only UK ancestry might determine if the signals are driven entirely by UK ancestry.

Conclusions: robustness, validity, reliability

Overall, I feel that the findings, particularly in relation to the 'new gene' are somewhat oversold and a bit forced into the paper.

Version 1:

Decision Letter:

24th January 2024

Dear Rafik,

Your revised Letter entitled "Large scale genome-wide association analyses identify novel genetic loci and mechanisms in hypertrophic cardiomyopathy" has been seen by two of the original referees. (Reviewer #3 was also invited to comment on the revision but has not submitted a review to date.) You will see from the comments below that, while Reviewers #1 and #2 find the study improved, Reviewer #2 has requested additional analyses. We remain interested in the possibility of publishing your study in Nature Genetics, but we would like to consider your response to these requests in the form of a further revision before we make a final decision on publication.

As before, to guide the scope of the revisions, the editors discuss the referee reports in detail within the team, including with the chief editor, with a view to identifying key priorities that should be addressed in revision, and sometimes overruling referee requests that are deemed beyond the scope of the current study. In this case, we agree with Reviewer #2 that it would be useful to extend the rare variant burden analyses to include all genes and to explore results obtained using a less stringent minor allele frequency filter. We also ask that you present the corresponding QQ plots for the rare variant burden analyses and make the full summary statistics available. We again hope you will find this prioritized set of referee points to be useful when revising your study. Please do not hesitate to get in touch if you would like to discuss these issues further.

We invite you to revise your manuscript taking into account all reviewer and editor comments. Please highlight all changes in the manuscript text file. At this stage, we will need you to upload a copy of the manuscript in MS Word .docx or similar editable format.

*2) If you have not done so already please begin to revise your manuscript so that it conforms to our Letter format instructions, available

[here](http://www.nature.com/ng/authors/article_types/index.html).

*3) Include a revised version of any required Reporting Summary: <https://www.nature.com/documents/nr-reporting-summary.pdf>

Link Redacted

We hope to receive your revised manuscript within 4-8 weeks. If you cannot send it within this time, please let us know.

Nature Genetics is committed to improving transparency in authorship. As part of our efforts in this direction, we are now requesting that all authors identified as 'corresponding author' on published papers create and link their Open Researcher and Contributor Identifier (ORCID) with their account on the Manuscript Tracking System (MTS), prior to acceptance. ORCID helps the scientific community achieve unambiguous attribution of all scholarly contributions. You can create and link your ORCID from the home page of the MTS by clicking on 'Modify my Springer Nature account'. For more information, please visit www.springernature.com/orcid.

Sincerely,

Kyle

Kyle Vogan, PhD
Senior Editor
Nature Genetics
<https://orcid.org/0000-0001-9565-9665>

Referee expertise:

Referee #1: Genetics, cardiovascular diseases, cardiac MRI

Referee #2: Genetics, cardiovascular diseases, statistical methods

Referee #3: Genetics, cardiovascular diseases, cardiomyopathies

Reviewers' Comments:

Reviewer #1:
Remarks to the Author:

Thank you to the authors for their extensive work to respond to my remarks and the other reviewers. I am satisfied with the responses to my questions and the additional information added to the paper, new figure 4 works well and great to have additional reporting on results from the MTAG approach. The inclusion of testing for rare LoF variants across 26 candidate genes is a good addition to the paper. My only remaining comment concerns you may wish to revise the abstract to demonstrate additional work done on rare variant testing on candidate genes?

Reviewer #2:
Remarks to the Author:

The authors' response has addressed many of the reviewers' concerns. There are still some remaining questions that were not addressed.

The authors extended the burden analysis to 26 genes rather than looking at the whole exome to maintain statistical power. I do not think this is sufficient and I think the authors should test all genes for associations in a burden test. They can still test the 26 genes in a more focused hypothesis and adjust only for the 26 genes tested. It is important that the authors provide QQ plots for the burden analysis for quality control and I cannot see any reason for not doing so. The negative control analysis is not sufficient. Furthermore, I think the authors should make the summary level data from the burden analysis publicly available to other researchers as is now standard to do with GWAS data.

The authors claim that it is standard and necessary to use a $MAF < 10^{-4}$ filter in the analysis and state that no HCM above that frequency has been reported. I do not agree with that. There are e.g. founder mutations in MYBPC3 that have a much higher frequency and cause HCM. I think all predicted loss-of-function variants in the genes should be explored. If there are more frequent variants, e.g. in SVIL, that do not associate with HCM and render the burden association non-significant, that should then be discussed.

The authors state that exploring a bi-directional relationship between HCM and LV contractility is outside of the scope of the manuscript. Confounding due to reverse causation, however, can complicate the interpretation of MR findings. I feel that it needs to be explored and potentially bi-directional relationships need to be accounted for in MR analysis.

Reviewer #3:
None

Version 2:

Decision Letter:

Our ref: NG-LE61789R1

9th May 2024

Dear Rafik,

Thank you for submitting your revised manuscript "Large scale genome-wide association analyses identify novel genetic loci and mechanisms in hypertrophic cardiomyopathy" (NG-LE61789R1). In light of your responses to the points raised at the previous round of review, we will be happy in principle to publish your study in Nature Genetics as a Letter pending final revisions to comply with our editorial and formatting guidelines.

We are now performing detailed checks on your paper, and we will send you a checklist detailing our editorial and formatting requirements soon. Please do not upload the final materials or make any revisions until you receive this additional information from us.

Thank you again for your interest in Nature Genetics. Please do not hesitate to contact me if you have any questions.

Sincerely,
Kyle

Kyle Vogan, PhD
Senior Editor
Nature Genetics
<https://orcid.org/0000-0001-9565-9665>

Version 3:

Decision Letter:

In reply please quote: NG-LE61789R2 Tadros

10th January 2025

Dear Rafik,

I am delighted to say that your manuscript "Large scale genome-wide association analyses identify novel genetic loci and mechanisms in hypertrophic cardiomyopathy" has been accepted for publication in an upcoming issue of Nature Genetics.

Your paper will be published online after we receive your corrections and will appear in print in the next available issue. You can find out your date of online publication by contacting the Nature Press Office (press@nature.com) after sending your e-proof corrections.

Before your paper is published online, we will be distributing a press release to news organizations worldwide, which may very well include details of your work. We are happy for your institution or funding agency to prepare its own press release, but it must mention the embargo date and Nature Genetics. Our Press Office may contact you closer to the time of publication, but if you or your Press Office have any enquiries in the meantime, please contact press@nature.com.

Acceptance is conditional on the data in the manuscript not being published elsewhere, or announced in the print or

electronic media, until the embargo/publication date. These restrictions are not intended to deter you from presenting your data at academic meetings and conferences, but any enquiries from the media about papers not yet scheduled for publication should be referred to us.

Authors may need to take specific actions to achieve [compliance](https://www.springernature.com/gp/open-research/funding/policy-compliance-faqs) with funder and institutional open access mandates. If your research is supported by a funder that requires immediate open access (e.g. according to [Plan S principles](https://www.springernature.com/gp/open-research/plan-s-compliance)) then you should select the gold OA route, and we will direct you to the compliant route where possible. For authors selecting the subscription publication route, the journal's standard licensing terms will need to be accepted, including [those licensing terms](https://www.nature.com/nature-portfolio/editorial-policies/self-archiving-and-license-to-publish) will supersede any other terms that the author or any third party may assert apply to any version of the manuscript.

If you have not already done so, we strongly recommend that you upload the step-by-step protocols used in this manuscript to protocols.io. protocols.io is an open online resource that allows researchers to share their detailed experimental know-how. All uploaded protocols are made freely available and are assigned DOIs for ease of citation. Protocols can be linked to any publications in which they are used and will be linked to from your article. You can also establish a dedicated workspace to collect all your lab Protocols. By uploading your Protocols to protocols.io, you are enabling researchers to more readily reproduce or adapt the methodology you use, as well as increasing the visibility of your protocols and papers. Upload your Protocols at <https://protocols.io>. Further information can be found at <https://www.protocols.io/help/publish-articles>.

Sincerely,
Kyle

Kyle Vogan, PhD
Senior Editor
Nature Genetics
<https://orcid.org/0000-0001-9565-9665>

Click here if you would like to recommend Nature Genetics to your librarian
<http://www.nature.com/subscriptions/recommend.html#forms>

**Visit the Springer Nature Editorial and Publishing website at http://editorial-jobs.springernature.com?utm_source=ejP_NGen_email&utm_medium=ejP_NGen_email&utm_campaign=ejp_NGen for more information about our career opportunities. If you have any questions, please click [here](mailto:editorial.publishing.jobs@springernature.com).

NG-LE61789

Large scale genome-wide association analyses identify novel genetic loci and mechanisms in hypertrophic cardiomyopathy

Response to reviewers comments

We thank the Editors and Reviewers for the careful revision and for providing us with an opportunity to improve our manuscript. We hereby provide a point-by-point response in red to each Reviewer comment (copied in black). Changes in the manuscript or supporting tables and figures, if any, are described within each response. When appropriate, the text change is also copied in the response. All changes made are highlighted in yellow in the main manuscript text.

Editors' prioritized set of referee points:

- 1) Address all technical queries related to the association analyses, particularly with respect to
 - a. assessing heterogeneity across the study cohorts, and
 - b. interpreting the MTAG analyses
- 2) Extend the burden analyses to encompass a larger set of genes, if feasible

We thank the editors for providing a prioritized set of referee points, which we address within our response to the reviewers.

Reviewer #1:

Summary of paper

This paper describes the results from a GWAS of HCM (5000 HCM cases, 68,000 controls) and MTAG analysis including over 5000 HCM cases, 68,000 controls and c. 36,000 individuals with cardiac MRI data. The authors report 70 loci (50 novel) for HCM, 62 (32 novel) for LV traits measured using cardiac MRI. One novel gene is highlighted and discussed – supervillin. MR analysis exploring hypertension and LV contractility supports a causative role in both obstructive and non-obstructive forms of HCM.

Comments to authors

Increasing sample size from prior work by the co-authors with new cohorts has yielded many new loci from both GWAS of all HCM patients, sarcomere positive and negative subsets, and new additional loci using MTAG analysis.

There are several suggestions and points of clarification I would like to make:

1. GWAS of HCM – selection and information on the cases and controls – how was matching done and analysis? ST1 does not provide information on the controls? Can this information be included and ancestry matching, what are the demographics of the controls?

A detailed description of the 7 included case-control datasets including matching appears in the Supplementary Note, with reference to prior publications for datasets included in previous clinical or genomics studies (PMIDs 33495597, 31699273, 33495596, 27869827, 22184326, 32581362). As suggested by the reviewer, we now added demographic data for the cases and controls in Supplementary Table 1, including age and genetic ancestry. As routinely performed in GWAS, we accounted for residual ancestral differences in cases and controls by correcting for genotypic principal components. With proportional matching case/control selection and with genetic ancestry included in the association testing, we do not expect heterogeneity in our results due to ancestry mismatch. The genomic inflation factors (λ_{GC}) for each dataset also appear in Supplementary Table 1 and range from 0.99 to 1.05.

2. The MTAG method has been used to leverage additional loci associated with HCM. This is a method used in prior work by some of the authors for identifying HCM loci. This method provides loci which are important to validate. I wondered as this was a method used in prior work – how many of the loci discovered from the MTAG analysis in their prior study were genome-wide significant in this new meta analysis of HCM? The results from this would provide validation of this method for leveraging loci from this method for HCM and thus confidence in this new number of loci now being reported. Can these results be reported and discussed?

We thank the reviewer for their suggestion. In our previous study (Tadros et al, <https://doi.org/10.1038/s41588-020-00762-2>), we indeed performed multitrait analysis of GWAS (MTAG) for HCM (and DCM) leveraging the strong genetic correlations between left ventricular (LV) traits and cardiomyopathy. MTAG allowed us to uncover 10 additional HCM loci at the time. In this previous study, all 10 MTAG loci were validated in an independent dataset, highlighting the

robustness of the approach. These 10 previously reported MTAG loci all reach $P < 5 \times 10^{-8}$ in the present GWAS, again highlighting the robustness of HCM-MTAG when leveraging genetically correlated LV traits. This is now highlighted in the main text, as follows: “a prior study strongly supports the robustness of the HCM-LV traits MTAG approach, whereby all 10 HCM loci uncovered using MTAG in this previous study were independently validated, and all reach $P < 5 \times 10^{-8}$ in the present GWAS (Table 1).”

3. How reliable are MTAG estimates of effect sizes?

To assess the reliability of MTAG estimates of effect sizes, we compared the regression coefficients derived from the GWAS and those computed with MTAG, for 1) all SNPs, and 2) for SNPs reaching $P < 0.001$ in either GWAS and/or MTAG. Effect estimates strongly correlated, with Spearman’s correlation coefficients of 0.90 and 0.96, and slopes of 0.83 and 1.00, respectively. This analysis is now presented as a new Supplementary Figure 13 (copied below), and briefly mentioned in the manuscript text as follows: “Effect estimates derived from MTAG were strongly correlated with those from GWAS both overall and for variants associated with HCM at $P < 0.001$ in either MTAG and/or GWAS (Spearman’s correlation coefficient 0.96; regression slope 1.00; Supplementary Figure 13)”.

4. I was intrigued with the discovery and discussion of supervillin – indeed it is an interesting candidate but it was not genome-wide significant for all HCM, noting it was for MTAG (add this information to Figure 1). Follow up analyses provides support for this as a candidate gene for HCM and nicely demonstrates leveraging related traits. As there is no validation for other loci – is this an approach which you can use for potential validation of other interesting candidate genes from this work? This would add value to the manuscript as readers would be interested in knowing which genes may have rare variants and the data looks to be available if done for supervillin?

We thank the reviewer for finding our *SVIL* result and approach interesting. We have now extended our rare variant burden analysis to 26 genes that were prioritized based on concordance by both

OpenTargets and FUMA, as well as LV specific expression in bulk RNAseq data (GTEx v8) and expression in cardiomyocytes using publicly available snRNA-seq data (previous Supplementary Figure 13). We performed targeted rare variant association for this set of 26 genes rather than the whole exome to reduce the multiple testing burden and hence maintain power. We included case-control datasets from Genomics England (GeL), BRRD and UK Biobank. For *SVIL*, HCM cases were also included from the Oxford Medical Genetics Laboratory (because this gene had been added to the diagnostic gene panel previously). The results are now presented in a new Figure 4 (copied below) which extends on prior Supplementary Figure 13 and prior Figure 2. Detailed results of single cohorts for all 26 genes are now presented in Supplementary Table 16. An updated list of *SVIL* truncating variants (now including UK Biobank) appears in Figure 4C, and Supplementary Table 17.

The text was adjusted as follows : “GWAS loci often co-localize with genes harboring disease-causing rare variants.²⁶ To identify novel HCM disease genes, we explored whether rare ($MAF < 10^{-4}$) predicted loss-of-function (LoF) variants in the 26 prioritized genes from significant GWAS/MTAG loci are associated with HCM. We performed case-control burden testing using sequencing data from Bioresource of Rare Diseases (BRRD), Genomics England (GEL), UKB, and the Oxford Medical Genetics Laboratory (OMGL; only for *SVIL*), followed by a fixed-effects model inverse variance weighted meta-analysis comprised of up to 2,502 unrelated HCM cases and 486,217 controls (Figure 4B and Supplementary Table 16A). Rare LoF variants in *ALPK3* and *SVIL* were significantly associated with HCM at the Bonferroni-corrected $P < 0.0019$ ($0.05/26$). While truncating variants in *ALPK3* have previously been shown to cause HCM and are now included in most clinical testing panels, *SVIL* represents a novel HCM gene with a comparable effect size for LoF variants (OR 10.5, 95% CI 4.3-26.1; $P = 3.6 \times 10^{-7}$). The effect estimates for *SVIL* LoF variants do not show significant heterogeneity across the 4 datasets (Supplementary Figure 16) and the associations remain significant when excluding each dataset one at a time (Supplementary Table 16B). Furthermore, synonymous variant burden testing was performed as a negative control and does not show significant associations (Figure 4B and Supplementary Table 16C). *SVIL* LoF variants found in 8 unrelated cases are listed in Supplementary Table 17 and Figure 4C. None of the 8 unrelated HCM cases that carry a *SVIL* LoF variant carries any other pathogenic or likely pathogenic variant.”

Last, as suggested by the reviewer, we added the significance of the *SVIL* locus in MTAG to Figure 1.

5. The all sample GWAS for HCM provided 50 novel loci – the authors do not provide a detailed follow up discussion of all potential candidate genes at these loci – can any further work be done and reported on? I note a lot of emphasis on supervillin from the pairwise GWAS as an interesting gene but not so much discussion on other genes from this work?

We think that discussing every locus is beyond the scope of the current paper. In the main text, we briefly discuss the 26 prioritized genes mapped to HCM loci.

6. I was interested in the results from the GWAS of sarcomere (-) patients – 24 genome-wide significant loci are listed – opposed to the 70 in the full cohort. What is the overlap of loci are the same loci picked up – can you perform pathway analyses – do the results differ to results with all HCM patients?

Out of the 24 SARC- loci, 23 reached $P < 5 \times 10^{-8}$ in either the all-comer GWAS or MTAG. The genetic correlation (r_g) between SARC- GWAS and all-comer HCM GWAS was 0.98 (SE: 0.01; $P < 10^{-50}$). As the 2 are highly correlated with almost complete overlap in genetic signals, we suspect that pathway analysis will not reveal anything different.

We performed a lookup of the 68 MTAG loci in the SARC- and SARC+ analyses, and now report this in Supplementary Table 8. All the 68 loci reach nominal significance in the SARC- analysis except the *MYBPC3* locus (as expected since it tags rare *MYBPC3* variants causing SARC+ HCM, not SARC- HCM).

7. From your work what is new beyond the prior GWAS on the biology of HCM?

The present work identified 50 novel loci associated with HCM, a substantial increase in the understanding of the genetic basis of HCM. We also identify a novel gene (*SVIL*) harboring rare pathogenic truncating variants that cause HCM. There is need of course for follow-up functional studies to dissect the biological mechanisms underlying the 70 HCM loci, which is beyond the scope of the present genetics study. Such future studies may shed light on novel biology of HCM which can in turn lead to therapeutic innovations.

8. Are potential candidate genes at loci also found for other cardiovascular traits? Is there pleiotropy? This is not explored or discussed?

We thank the reviewer for the suggestion. We used OpenTargets to explore association of the 70 lead SNPs (or any other SNP in linkage disequilibrium, $R^2 > 0.5$) with any published cardiovascular/ cardiometabolic or other traits. Of the 70 loci associated with HCM, 51 were previously associated with cardiovascular and/or cardiometabolic traits, including traits related to ECG measures (37 loci), body mass (16), blood pressure (15), atrial fibrillation (9), left ventricular structure/function (8), atherosclerotic cardiovascular disease (8), and lipids (7). The full results of this prior GWAS lookup are now in a new Supplementary Table 15.

9. You report new loci from the study of cardiac MRI traits – these results are not discussed or put in context with others work on genetic analysis of cardiac MRI traits? Can this be commented upon?

The primary objective of the present study is the identification of genomic loci associated with HCM. The GWAS of LV traits was completed to allow for MTAG and for the MR analysis of LV contractility (see updated flowchart in Figure 1). All cardiac MRI traits loci are listed in Supplementary Table 6, together with data on gene mapping using OpenTargets and FUMA, and a lookup in the HCM GWAS. In the same table, we identify loci that have been previously reported in association with any LV trait either using MRI or cardiac ultrasound data (PMIDs 19584346, 28394258, 29403010, 31554410, 32382064, 32605384, 32814899, 33495596, 35278270, 35479509, 35697867, 36598836, 36944631, 37126556). A recent study by Pirruccello et al (PMID 35697867) explored in depth genetic overlap between left- and right-sided cardiac traits as well as main vessels. We now also report lookups of the 62 loci associated with LV traits (present GWAS) in the study of Pirruccello et al (Supplementary Table 6).

10. With MTAG leveraging loci for HCM – would it be fair to say the loci discovered from this analysis would also be loci for the cardiac MRI trait used as well – so the loci represent a dual phenotype, or am I misunderstanding this method? To rephrase my question – from the MTAG analysis reported previously by some of the co-authors, are loci also significant now in a GWAS of the cardiac MRI trait which was used in the MTAG?

MTAG leverages genetically correlated traits to increase statistical power to identify trait-specific genetic associations. In the present use of MTAG, we leveraged the strong genetic correlations between HCM and LV traits to identify additional loci associated with HCM. Although the MTAG output for HCM+LV traits also includes MTAG summary statistics for the included LV traits, we only considered and reported the summary statistics for the HCM-MTAG. In both our previous publication (Tadros et al, PMID 33495596) and the present study, we report the association analyses for LV traits of all HCM loci (GWAS and MTAG) (Supplementary Table 8). Concerning the 10 HCM loci identified

with MTAG in our prior study, all reached $P < 5 \times 10^{-8}$ in the present HCM GWAS (as described in response to comment #2 above). Since these are HCM-specific loci, not all reach $P < 5 \times 10^{-8}$ in the present LV trait GWAS, as described in the table below.

snpid	chr	bpos	a1	a2	strain_circ_z	strain_circ_p	LVconc_z	LVconc_p	LVESVi_z	LVESVi_p
rs9647379	3	171785168	G	C	2.04	2.7E-02	-5.61	1.0E-08	2.76	3.4E-03
rs2191445	5	57011469	A	T	5.15	5.3E-07	-3.08	1.8E-03	4.97	1.3E-06
rs4385202	5	138743256	A	G	-6.48	2.6E-10	2.08	5.5E-02	-5.21	2.6E-07
rs66761782	6	36636080	T	C	7.99	5.1E-17	-8.10	4.0E-16	4.78	9.8E-07
rs60871386	7	128430437	G	T	5.23	1.1E-07	-1.81	8.4E-02	5.95	1.7E-09
rs3740293	10	75406141	A	C	3.28	1.0E-03	-4.74	1.6E-06	0.31	8.6E-01
rs11196078	10	114487812	G	A	4.37	1.9E-05	-4.53	9.1E-06	1.83	6.2E-02
rs11073729	15	85350081	T	C	5.87	6.6E-09	-10.37	2.5E-24	5.53	4.4E-08
rs9892651	17	64303793	C	T	5.73	3.4E-09	-2.86	4.0E-03	5.72	4.2E-09
rs2186370	22	24171305	A	G	-7.32	4.3E-13	8.95	8.4E-19	-7.73	7.6E-15

11. You have performed slightly different downstream analysis from your different GWAS findings, but can your approach be commented on in the text – why was one pathway not followed?

In Figure 1, we have outlined the flow of our analyses. The stratified SARC+ and SARC- analyses showed high genetic correlation and yielded very few differences so we focussed on the more powered all-comer HCM disease phenotype. LV trait GWAS were performed to be used for MTAG and MR analyses only (as described in response to comment #9 above). As highlighted in our response to comment #4 above, we have extended our rare variant LoF burden analysis to a prioritized set of 26 genes and have updated the text and reported the findings in Supplementary Table 16 and Figure 4.

12. The MR work was interesting but I was a little uncertain of how this fit in with this paper. It seemed like an analysis which was tagged on and perhaps be the topic of a different paper?

The MR work follows on previous findings that increased LV contractility and blood pressure are causally associated with HCM. Previous studies were however not powered to explore causal associations for obstructive HCM (oHCM) and non-obstructive HCM (nHCM), separately. In clinical practice, the management of oHCM and nHCM is generally distinct. In oHCM, myosin inhibitors have been shown to be effective in reducing symptoms, but their use in nHCM remains uncertain. We leveraged the sample size of the current HCM cohort to address the timely question of whether myosin inhibition (and blood pressure control) may be potential therapeutic targets in both oHCM and nHCM, using two-sample MR. We are glad the reviewer finds this work interesting. While we considered the reviewer's suggestion to consider this work for another paper, we decided to keep it in this current paper as the results are considered timely at this stage. Randomized trials of myosin inhibitors in nHCM are being initiated, and there is now much attention on afterload reduction therapy (blood pressure control and/or obstruction relief) in nHCM and oHCM.

13. Are you able to perform genetic analysis of the obstructive and non-obstructive HCM into the paper if you then shift focus to this phenotype?

We performed case-control GWAS for oHCM and nHCM separately for the MR analysis, as described

in the methods section. These GWAS show strong genetic correlation ($r_g=0.87$, $SE=0.13$, $P=4 \times 10^{-11}$), reflecting shared genetic basis of oHCM and nHCM. We now report loci reaching $P < 5 \times 10^{-8}$ in oHCM and/or nHCM in Supplementary Table 2. All such loci reached significance in the all-comer HCM GWAS and/or MTAG, except one locus associated with nHCM (lead SNP rs56224968) mapped to *ACSL1*, a gene involved in fatty acid metabolism and whose biological association with nHCM remains uncertain. We also report lookup of the 68 MTAG loci lead SNPs in oHCM and nHCM in Supplementary Table 8. Unsurprisingly, all HCM risk alleles in the lead 68 variants show concordant effects in both nHCM and oHCM. Of the 68 variants, 24 and 39 reach nominal statistical significance ($P < 0.001$) in oHCM and nHCM, respectively.

Reviewer #2:

The work by Tadros et al. is a large meta-analysis of hypertrophic cardiomyopathy (HCM) where the authors utilize GWASs from 7 cohorts (5,900 HCM cases) and perform GWASs of cardiac magnetic resonance (CMR) traits ($N \sim 36,000$). They claim identification of 50 novel HCM loci and 32 novel loci associated with relevant left ventricular (LV) structural or functional traits. The authors claim SVIL as a novel HCM gene and explore HCM association with rare truncating variants in SVIL. Mendelian randomization analyses were conducted to explore the causal role of increased LV contractility in both obstructive (oHCM) and non-obstructive forms of HCM (nHCM). The work is well written and of interest, especially if the authors extend the Burden loss-of-function variant analysis beyond SVIL and perform it on all genes. This manuscript is somewhat a follow up of a paper published in 2021, a HCM GWAS utilizing $\sim 1,700$ HCM cases and LV traits ($N=19,999$) where they explored causal effects between increased LV contractility with HCM (not split into oHCM and nHCM).

Comments:

1) The authors use a genome-wide significance threshold of 5×10^{-8} . This threshold is somewhat outdated, and I would suggest using a Bonferroni threshold where the number of variants tested is accounted for, i.e. $0.05/9,492,702$. The authors do not have a replication set and they do not show the if the associations are consistent across the cohorts they are using. I understand that replication cohorts are difficult to come by, but then it is more important to explore heterogeneity between the cohorts and be conservative when selecting the GWS threshold.

The reviewer has rightly pointed out that the GWAS significance threshold used is the conventional 5×10^{-8} . The $\sim 9M$ variants tested in our study are not independent so a Bonferroni threshold will not be appropriate. Given that we are still exploring variants with $MAF > 1\%$ and the majority being highly correlated, we do not think a more stringent threshold is appropriate. This threshold is still used as standard in recent publications (PMIDs: 37248441, 37034649, 36474045) and, over the last 20 years, GWAS loci defined by this threshold have almost always been successfully replicated.

Apologies for the oversight on reporting heterogeneity statistics. We have now added Cochran's Q heterogeneity test P-values to Table 1 and Supplementary Table 8. We also provide forest plots in the supplement (Supplementary Figures 1 and 14) which show that all our loci are consistent across the cohorts, with the exception of a locus on chromosome 11 (lead SNPs rs78310129 in GWAS and rs182427065 in MTAG) which tags founder HCM-causing rare variants in *MYBPC3* (mostly in the Netherlands population; see PMIDs 33495596 and 33495597).

2) The authors perform a co-localization analysis searching for variants that associate with HCM and DCM with opposing genomic effects. They find an intergenic variant that associates suggestively (not genome-wide significant for HCM nor DCM) with HCM and DCM with opposing effects. In the analysis they require (somewhat arbitrarily) that the variants have a $P < 1 \times 10^{-5}$ for both phenotypes and have a posterior probability > 0.8 of colocalization. The authors acknowledge that this requires further evidence to support implication in HCM. This common variant association would then ideally be replicated in another HCM cohort. Somewhat surprisingly, the authors explore the closest gene (SVIL) in a burden analysis but this isn't a strict replication of the suggestive association. Did the common

variant close to *SVIL* show consistent HCM association across cohorts?

Although the *SVIL* locus and gene have initially attracted our attention in the context of the result of GWAS-PW, in light of reviewers' suggestions to extend the rare variant testing, we have now performed a systematic rare variant association including a set of 26 prioritized genes. See our response to Reviewer 1 (comment 4), and our response below to this reviewer (comment 3).

Nevertheless, we hereby clarify some points raised by the reviewer in their comment.

We used OpenTargets and FUMA, rather than proximity, to systematically assign a variant to gene(s) for all the discovered loci. For this locus, eQTL and sQTL data suggest rs6481586 acts via regulating *SVIL* (Figure A).

Figure A: OpenTarget (v22.10) result for rs6481586 showing *SVIL* with largest variant to gene (V2G) score while *LYZL1* is the nearest gene based on TSS from the lead SNP.

The common variant association with HCM is consistent across cohorts, as shown in the forest plot (Figure B), and supported by the Cochran's Q heterogeneity test P-value = 0.903.

Figure B: Forest plot for rs6481586 which is the lead SNP in this locus from all-comer meta-analysis. Since this SNP does not reach genome-wide significance in the GWAS ($P < 5 \times 10^{-8}$), this forest plot is not in Supplementary Figure 1.

Note that the *SVIL* locus reaches significance ($P = 1.1 \times 10^{-8}$) in the present HCM MTAG, and also reached significance ($P = 1.6 \times 10^{-9}$) in our previously published DCM MTAG (PMID 33495596).

We now provide the *SVIL* locus forest plot in the supplement, along with all forest plots for GWAS and MTAG loci (Supplementary Figures 1 and 14).

3) The loss of function burden analysis suggests that *SVIL* loss-of-function variants cause HCM. The

associations would survive a multiple testing adjustment where the number of genes in the genome would be accounted for. I would like to see this analysis not restricted to *SVIL*. I am sure that other genes in this analysis would be of interest and this would also help putting the *SVIL* association in perspective. Furthermore, the authors could provide QQ-plots etc. for QC analysis. They need to confirm that there is no inflation in test statistic etc. in the burden analysis.

As pointed out in our response to comment 4 of reviewer 1, we have extended our burden testing for the 26 prioritized genes mapped to GWAS/MTAG loci, including data from UKB, BRRD and GEL. We decided to perform burden testing for 26 genes instead of the whole-exome to maintain statistical power while accounting for multiple testing. Results are now discussed in the main text and presented in new Figure 4 (reproduced in our response to comment 4 of reviewer 1). Detailed results of burden testing are shown in Supplementary Table 16. In total, 2 of the 26 genes (including *SVIL*) show evidence of association with HCM at the Bonferroni-corrected $P < 0.0019$. The effect estimates for *SVIL* LoF variants do not show significant heterogeneity across the 4 datasets (Supplementary Figure 16) and the associations remain significant when excluding each dataset one at a time (Supplementary Table 16B).

As a negative control, we also performed and report on burden testing of rare ($MAF \leq 0.0001$) synonymous variants in the 26 prioritized genes using an otherwise identical analysis approach to the LoF variant analysis. Reassuringly, we do not observe significant associations of synonymous variants with HCM in any of the 26 tested genes.

4) Did the HCM cases that carried *SVIL* LoF variants also carry other pathogenic variants?

None of the *SVIL* LoF variant carriers carried another pathogenic or likely pathogenic variant, and this is now added to the manuscript main text.

5) The authors utilize data from the UKB in the manuscript but not in the burden analysis. Several individuals carry *SVIL* loss-of-function variants in the UKB and HCM diagnostic ICD codes and imaging data are available. Is there any reason not to explore that?

Thank you for suggesting the use of UKB for burden analysis. As pointed out in our response to comment 4 of reviewer 1, and to your comment 3 above, we have updated the manuscript incorporating meta-analysis of LoF burden analysis using UKB, BRRD and GEL.

6) Why use only variants with a $MAF < 10^{-4}$ in the non-Finnish European ancestral group of gnomAD v2.1.1? Are there other predicted loss-of-function variants found in the data that are more common?

We consider that it is standard, and necessary, to use a $MAF < 10^{-4}$ filter in analyses of potential novel HCM-causing rare variants as no proven HCM-causing variant exceeds this allele frequency threshold. Using an appropriate filter is needed to improve signal to noise.

7) Is there any available data to explore if the *SVIL* LoF heterozygotes have myofibrillar myopathy as described in OMIM for homozygotes?

As far as we can tell, none of the *SVIL* LOF heterozygotes have reported any skeletal muscle symptoms and so none have had muscle biopsy.

8) The authors say that they “performed rare variant burden analysis including 1,845 clinically diagnosed unrelated HCM cases”. Then they state “In one family, the *SVIL* LoF variant (p.(Gln255*)) was carried by two cousins with HCM”. This is confusing and needs to be clarified.

Only unrelated HCM probands were included in the burden testing. Following identification of *SVIL* LOF variants associated with HCM in these probands, co-segregation analyses were performed for families with multiple affected individuals. The statement is now clarified and updated in the manuscript, as follows: “Family screening provided limited evidence of co-segregation. In one family, variant *SVIL*:p.(Gln255Ter) was carried by two cousins with HCM, and in another family, variant *SVIL*:p.(Arg1616Ter) was carried by two siblings with HCM.”

9) The authors of the MTAG paper suggest using their method when the genetic correlation between traits is >0.7 and state that “Replication is the best way to assess the credibility of individual-SNP associations.” For HCM the traits have correlation <0.7 and there is no replication cohort available. The authors do perform an estimation of the upper bound for the FDR, but this analysis holds under certain assumptions. Do these assumptions hold?

For MTAG to perform well, the main assumption of the method is that the included SNPs share a variance covariance matrix of effect sizes across traits. In order to assess if this assumption is met, the MTAG authors performed simulations using differently powered pairwise studies with different genetic correlations and tested resulting maxFDR estimates (Suppl Figure 2 from Turley et al, PMID 29292387, copied below). MaxFDR scales with poor genetic correlations among the included traits or high imbalance in the statistical power of the included summary statistics as measured by χ^2 . The 3 LV traits used in our analysis have genetic correlations ≥ 0.48 with HCM. Furthermore, the mean χ^2 for the 4 traits used for our MTAG analysis are very similar (range 1.134-1.181, now added to Figure 2). In our MTAG analysis, based on the simulations performed by the MTAG authors (PMID 29292387), the expected maxFDR would be ~ 0.05 (from figure below) while we estimated it to be 0.027. The reviewers have rightly pointed out that the recommendation is to replicate the MTAG findings. As described earlier for reviewer 1, we have shown that all 10 MTAG loci published in Tadros et al (2021) were independently replicated in Tadros et al (2021) and reached $P < 5 \times 10^{-8}$ in our current GWAS meta-analysis, supporting the robustness of our HCM MTAG findings.

Supplementary Figure 2. Two-trait illustration of MTAG’s maxFDR. (a) The y-axis is the maxFDR attainable under certain assumptions, including the assumption that the fraction of SNPs that are non-null for any individual trait is at least 10%, holding fixed r_B , r_ϵ , expected χ_1^2 , and expected χ_2^2 (see **Online Methods**). In this panel, $r_\epsilon = 0$ and expected $\chi_1^2 = 1.1$. (b) The y-axis is maxFDR as in panel (a), but the expected χ^2 -statistic is constrained to be equal for the GWASs of the two traits.

10) For the MTAG analysis, I would again suggest using a more conservative GWS threshold and show the HCM associations across all cohorts. Could the FinnGen dataset or UKB be useful?

A maximal false-discovery rate (maxFDR) estimator is included in the MTAG software. MaxFDR estimates the upper bound of false discovery. In our analysis, this maxFDR analysis shows that our MTAG results have a 2.7% rate of false discovery, so the 5×10^{-8} threshold is already conservative.

We had considered including FinnGen (Release 4: 432 HCM cases) and UK Biobank (analysis done in 2019: 325 HCM cases) for our all-comer HCM meta-analysis as the summary statistics were available at the time. We performed an M-statistics analysis (PMID: 28459806) that identifies studies that are systematically stronger or weaker than average and found both FinnGen and UK Biobank to be weaker than the others. Hence, we did not include them in our meta-analysis.

Figure A: M-statistic scatter plot against the average effect size of the 28 instruments used from Harper et al for this analysis. Dashed line indicates the Bonferroni corrected 5% significance threshold to allow multiple testing of 8 studies.

We have re-analysed UK Biobank now with the updated cases (674 HCM cases) and also downloaded the most recent FinnGen release (R9: 1044 HCM cases) and performed M-statistics again. Figure B below shows that FinnGen R9 is still under-powered. However, UK Biobank (UKB_23) is performing similarly to some other cohorts.

Figure B: M-statistic scatter plot against the average effect size of the 28 instruments used from Harper et al for this analysis. Dashed line indicates the Bonferroni corrected 5% significance threshold to allow multiple testing of 8 studies.

Including UK Biobank into our existing meta-analysis at this stage would impact on all the downstream analyses and results. We are in the process of collating more HCM case/control cohorts with a view to doing a further meta-analysis when we have a significant step up in numbers, so we feel it makes more sense to bundle UKB data at that point.

With regards to replication, Skol et al (PMID 16415888) suggest that replication based analysis is less powered if the proportion of samples in stage 1 is greater than 30%. Given that our stage 1 has >90% samples, we do not expect that UK Biobank results are adequately powered for replication purposes. We have performed a lookup of UK Biobank results for the 68 MTAG loci and, as expected, we only see 4 out of 68 with $P < 0.0007$ ($0.05/68$ tests). The 4 loci that replicate in the UK Biobank are *HSPB7* (rs1048302; $P = 2 \times 10^{-5}$), *CDKN1A* (rs3176326; $P = 7 \times 10^{-7}$), *BAG3* (rs17617337; 2×10^{-4}), and *FHOD3* (rs2644262; $P = 2 \times 10^{-5}$).

As suggested by the reviewer, we now show association data of all 68 MTAG loci for all HCM case-control cohorts, including Cochran's Q heterogeneity P-value in Supplementary Table 8, and Forest plots in Supplementary Figure 14. Of the 68 loci, only 1 shows significant evidence of heterogeneity of effect estimates across the HCM case-control cohorts: lead variant rs182427065 near *MYBPC3* which tags founder *MYBPC3* variants most prevalent in the Netherlands population (see PMIDs 33495596 and 33495597).

11) The Mendelian randomization analysis is interesting and seems to be well conducted. There are however variants deviating from the regression line, e.g. variants that associate with LVEF but have no effect on oHCM or nHCM. Doesn't this question the claim of a causal relationship? Is there a reason to think that those instruments that deviate are invalid?

We thank the reviewer for their appreciation of the MR analyses. Variants deviating from the MR regression line can reflect horizontal pleiotropy. To further explore such pleiotropy in our analysis and to evaluate the consequence of excluding outlier SNPs, we used the MR pleiotropy residual sum and outlier (MR-PRESSO) analysis (PMID 29686387). MR-PRESSO consists of 3 steps: testing for horizontal pleiotropy (global test), correcting for horizontal pleiotropy using outlier removal (outlier test) and evaluating differences in the causal estimate before and after outlier removal (distortion test). The results are added to Supplementary Table 19. In brief, all HCM ~ contractility MR analyses show significant horizontal pleiotropy (global test). After correction for horizontal pleiotropy using the outlier test, HCM ~ contractility MR analyses that were previously significant remain significant (i.e. all except those using the global longitudinal strain). Furthermore, outlier correction does not significantly alter the causal estimates for 11 of the 12 analyses (distortion test). Only the nHCM ~ LVEF analysis shows a significant distortion test, although the difference in effect magnitude is physiologically minimal (uncorrected IVW OR=1.44 vs. outlier-corrected IVW OR=1.31; OR reported per SD increase in LVEF).

12) Doesn't the possibility of a bi-directional relationship between HCM and LV contractility need to be explored (also just HCM affecting LV contractility rather than vice versa)?

The MR analysis we performed is targeted towards a precise biologically plausible hypothesis. Specifically, we tested whether increased LV contractility is causally associated with HCM and its obstructive and non-obstructive subtypes. The rationale for this hypothesis stems from the

observation that HCM-causing rare sarcomeric variants result in increased contractility (e.g. PMID 26912705) and that myosin inhibitors prevent and reverse hypertrophy in mice models (PMID 26912705) and relieve symptoms in human obstructive HCM (PMID 32871100). The benefit of myosin inhibitors in human non-obstructive HCM remains uncertain, hence our MR analysis, pending ongoing phase 3 clinical trials with mavacamten (ODYSSEY-HCM) and aficamten (ACACIA-HCM).

MR analyses beyond this hypothesis are certainly of interest but beyond the scope of this work. Please note that Reviewer 1 found the MR analysis interesting but was uncertain about its inclusion in this manuscript. We therefore thought that reporting MR analyses beyond the above-mentioned specific biologically plausible (and timely) hypothesis would deviate the focus of the manuscript.

13) Why do the authors restrict the analysis to variants with MAF>1%?

The table below shows the genetic power in the 3 biggest cohorts used in this meta-analysis. There is very limited power in each cohort for low allele frequency variants.

	cases	controls	EAF=0.5%	EAF=1%	EAF=2%	EAF=3%
HCMR	2431	40283	0.029	0.198	0.728	0.954
CAN	1035	13889	0.002	0.017	0.119	0.316
NL	999	2117	0.001	0.008	0.06	0.17

Table A: Genetic power estimates using GAS power calculator (https://csg.sph.umich.edu/abecasis/gas_power_calculator/) based on prevalence of 1/500 and genetic risk of 1.7 at different allele frequencies.

Given we are meta-analyzing 7 studies, we kept results with MAF>=1% to balance signal-to-noise.

14) There are likely some issues with the conditional analysis being performed, e.g. variants that are on chromosome 11 and are significant in the analysis are likely correlated and I suspect that they may not be associating if the authors would use genotypes rather than summary level data (COJO) to perform the conditional analysis.

The reviewer is correct. The common variants on chromosome 11 tag founder causal rare variants in *MYBPC3*. We have looked into this in Harper et al (PMID: 33495597) where we modeled the impact of both rare and common variants using multiple logistic regression and confirmed that HCM risk could be entirely attributed to the rare variants, specifically p.Arg502Trp and p.Trp792ValfsTer41. To avoid confusion to the readers, we removed variants chr11:43656852-70419528 from Supplementary Table 3, and added a footnote to the Supplementary Table.

15) The authors state that a “A locus on chromosome 11 which includes MYBPC3, a well-established disease gene, is associated with HCM and HCMSARC+, but not HCMSARC-, implying that this association is tagging known founder pathogenic variants in MYBPC3”. Although this may be the case, I cannot see why these associations would imply that there is founder variant (A genetic alteration observed with high frequency in a group that is or was geographically or culturally isolated)? Do all variants that associate with HCMSARC+ and not HCMSARC- tag a founder variant?

This statement is based on data from our prior studies (PMIDs 33495596 and 33495597), rather than data provided in the present manuscript. To avoid any confusion, we now reformulated the sentence as follows : “A locus on chromosome 11 which includes *MYBPC3* is associated with HCM and HCM_{SARC+} (but not HCM_{SARC-}). This locus is comprised of common variants that have previously been shown to tag known founder pathogenic variants in *MYBPC3*.”

16) I suggest annotating the variants using Variant effect predictor (VEP). I would e.g. like to know if any of the variants in main Table 1 are missense or loss-of-function.

We thank the reviewer for this suggestion. We now identify protein-altering variants in Table 1. Specifically, 4 of the variants appearing in Table 1 are missense variants:

- *BAG3* (ENST00000369085.3): p.Cys151Arg
- *ADPRHL1* (ENST00000375418.3): p.Leu294Arg
- *RNF207* (ENST00000377939.4): p.Gly603Ala
- *ZFYVE26* (ENST00000347230.4): p.Ser1040Thr

There were no other protein-altering variants in Table 1.

Similarly, we provide the most severe consequence for lead variants from the MTAG in Supplementary Table 8. We also provide detailed VEP annotations for all GWAS and MTAG lead variants in Supplementary Table 11.

Reviewer #3:

Summary of the key results

This is a large genome-wide association study (GWAS) and multi-trait analysis (MTAG) comprising a number of HCM cohorts (5,900 cases, 68,359 controls) and 36,083 UK Biobank (UKB) participants with cardiac magnetic resonance (CMR) imaging. The authors identify a total of 70 loci (50 novel) traits associated with HCM, and 62 loci (32 novel) associated with left ventricular (LV) structural or functional traits. Prominence is given to, *SVIL*, which encodes the actin-binding protein supervillin, making the suggestion that rare truncating *SVIL* variants cause HCM.

Originality and significance

There are already a number of GWAS studies in this area, including one by some of the same authors:

- Harper AR, Goel A, Grace C, Thomson KL, Petersen SE, Xu X, et al. Common genetic variants and modifiable risk factors underpin hypertrophic cardiomyopathy susceptibility and expressivity. *Nat Genet.* 2021;53(2):135-42.
- Tadros R, Francis C, Xu X, Vermeer AMC, Harper AR, Huurman R, et al. Shared genetic pathways contribute to risk of hypertrophic and dilated cardiomyopathies with opposite directions of effect. *Nat Genet.* 2021;53(2):128-34.
- Biddinger KJ, Jurgens SJ, Maamari D, Gaziano L, Choi SH, Morrill VN, et al. Rare and Common Genetic Variation Underlying the Risk of Hypertrophic Cardiomyopathy in a National Biobank. *JAMA Cardiol.* 2022;7(7):715-22.

The previous paper by Tadros has a similar design and conclusion. The only major difference is the size of the analysed cohort.

For an uncommon disease such as HCM, sample size is a major limitation to detect significant GWAS loci. The increase in power in this analysis compared to previous publications has helped us identify 70 loci compared to 20 previously known loci which is a major uplift in discovery.

Data & methodology

The meta-analysis for "single-trait" is robust with some interesting hits. However, the paper focuses very heavily on the "new gene" *SVIL* - based on data solely from the GWAS. The same locus did not reach significance with the single trait analysis and a case vs control burden test for rare variants.

The *SVIL* locus did not reach genome-wide significance in the single trait analysis (Table 1) but it became genome-wide significant in MTAG analysis (Figure 3, Supplementary Table 8). It was also significant in a prior DCM MTAG (PMID 33495596). We have extended the rare variant LOF burden analysis by including other genes and UK Biobank data. The association of *SVIL* LOF variants remains significant despite correction for multiple testing (odds ratio=10.5; P-value = 3.6×10^{-7}). We also confirm significant associations for *ALPK3* LOF variants, as shown in Supplementary Table 16 and Figure 4B (copied below). Association of synonymous variants with HCM was tested as a negative control and was not significant. The effect estimates for *SVIL* LoF variants do not show significant

heterogeneity across the 4 datasets (Supplementary Figure 16) and the associations remain significant when excluding each dataset one at a time (Supplementary Table 16B).

Supportive data on the significance of SVIL are few with little if any functional data (the only supporting information given is "cardiac abnormalities" in zebrafish).

Importantly, the authors mention a family where two cousins carry the variant, which is insufficient to support segregation. More data are required on this family and other patients in whom a Mendelian trait is genuinely suspected. As it stands, there is insufficient information to judge the veracity of the claim that SVIL is a new disease gene.

Within the familial data currently available we are only aware of two small, nuclear families where more than one individual with clinically diagnosed HCM carry an SVIL LOF variant: two siblings in one family and two cousins in another. We accept that we therefore have limited opportunity to test for co-segregation and limited evidence (albeit concordant in these two small families). That said, the ORs for SVIL LOFs (~10, with good reproducibility across cohorts and robust statistical support) are in a range that predicts familial disease rather than just a common variant susceptibility. For example they are comparable to those found for ALPK3 LOF alleles and ALPK3 is now accepted as a rare variant disease gene for familial HCM. Reflecting this we have changed wording in the text:

“ Rare LoF variants in *ALPK3* and *SVIL* were significantly associated with HCM at the Bonferroni-corrected $P < 0.0019$ ($0.05/26$). While truncating variants in *ALPK3* have previously been shown to cause HCM and are now included in most clinical testing panels, *SVIL* represents a novel HCM gene with a comparable effect size for LoF variants (OR 10.5, 95% CI 4.3-26.1; $P = 3.6 \times 10^{-7}$). The effect estimates for *SVIL* LoF variants do not show significant heterogeneity across the 4 datasets (Supplementary Figure 16) and the associations remain significant when excluding each dataset one at a time (Supplementary Table 16B). Furthermore, synonymous variant burden testing was performed as a negative control and does not show significant associations (Figure 4B and Supplementary Table 16C). *SVIL* LoF variants found in 8 unrelated cases are listed in Supplementary Table 17 and Figure 4C. None of the 8 unrelated HCM cases that carry a *SVIL* LoF variant carries any other pathogenic or likely pathogenic variant. Family screening provided limited evidence of co-segregation. In one family, variant *SVIL*:p.(Gln255Ter) was carried by two cousins with HCM, and in another family, variant *SVIL*:p.(Arg1616Ter) was carried by two siblings with HCM. *SVIL* encodes

supervillin, a large, multi-domain actin and myosin binding protein with multiple muscle and non-muscle isoforms, of which the muscle isoform has known roles in myofibril assembly and Z-disk attachment. *SVIL* is highly expressed in cardiac, skeletal, and smooth muscle myocytes in the Genotype Tissue Expression (GTEx) v9 snRNA-seq dataset, and *SVIL* morpholino knockdown in zebrafish produces cardiac abnormalities. In humans, loss of function (LoF) *SVIL* variants have been associated with smaller descending aortic diameter and homozygous LoF *SVIL* variants have been shown to cause a skeletal myopathy with mild cardiac features (left ventricular hypertrophy). Of interest, common variants in the *SVIL* locus are also associated with DCM, and using a Bayesian pairwise analysis approach (GWAS-PW), including the present HCM GWAS meta-analysis and a published DCM GWAS, we show that DCM and HCM share the same causal SNP but with the expected opposite directions of effect (Supplementary Table 18). Taken together, these data support *SVIL* as the likely causal gene in the HCM GWAS locus and identify *SVIL* as a novel disease gene for HCM, in which rare LoF alleles have an effect size similar to minor HCM disease genes tested in clinical practice. ”

As there is not much biological evidence for *SVIL*, it is possible that this is a false discovery due to the fact that *SVIL* is a locus for the imaging trait jointly under study (which is possible as *SVIL* was picked up as a candidate gene for RV traits in another CMR GWAS (<https://www.nature.com/articles/s41588-022-01083-2>)).

We consider that the reinforced burden test data are robust and these are not driven by the trait association, so do not believe that this is a false positive finding.

The HCM cohorts enrolled in this study are very diverse and are likely to be subject to considerable ascertainment bias. The data on cohort characteristics are very sparse and under “HCM physiology available” absent. The data should be expanded to include a full description including missingness of the analysed traits. A sensitivity analysis to examine the effect of cohort/centre would be helpful. Even in the few data provided, there is a curious anomaly of fairly advanced age, which is unusual for HCM.

In response to comment 1 of reviewer 1, we now added demographic data to Supplementary Table 1, including age and ancestry for cases and controls. Granular clinical data are unfortunately not available for some of the included case-control cohorts. Imaging data prior to septal reduction therapy or cardiac transplantation was available for a total of 4,630 (78%) of the overall study cohort, but 2,358 (97%) of the prospective HCMR cohort. Our heterogeneity analyses (Cochrane’s Q and forest plots) did not show significant heterogeneity across cohorts.

Regarding “HCM physiology”, we used a strict definition for obstructive (oHCM) vs. non-obstructive (nHCM) HCM to allow for robust stratified MR analyses addressing the question of whether increased LV contractility is causally associated with both nHCM and oHCM. Such a strict definition required clinical chart reviews and was only feasible for the 2 largest datasets (HCMR and the Canadian dataset). These 2 datasets allowed for sufficient power to demonstrate the causal association of increased contractility with nHCM and oHCM.

Regarding “fairly advanced age”, the average age of our study cohorts is very similar to contemporary HCM cohorts which are predominantly comprised of older patients with sarcomere-negative HCM.

See figure below reproduced from a study by SHaRE investigators (Canepa et al, PMID 32894986) showing that average age at diagnosis is increasing over time (51 ± 16 for patients diagnosed after 2010, with $\sim 11\%$ aged ≥ 70 years old at the time of diagnosis).

Appropriate use of statistics and treatment of uncertainties

As in a previous study by the same authors, the concept of the multi-trait analysis is a little difficult to understand. It is used to boost the power of the GWAS by doubling the number of loci through a combination of HCM-related loci with those that associate with functional and structural traits such as strain, ventricular volume and mass. While there may be overlap, I am not sure that we can say the latter are genuine HCM loci.

MTAG makes sense if the traits are closely correlated. For example a genotypic correlation > 0.7 . It is a little surprising to see the use of MTAG for a mix of all imaging traits plus HCM/DCM as it is likely the genotypic correlation will be weaker among these traits and type 1 error will be substantially inflated.

The key concept behind MTAG is that, when GWAS estimates from different traits are correlated, the effect estimates for each trait can be improved by appropriately incorporating information contained in the GWAS estimates for the other traits. In other words, MTAG is not simply a meta-analysis of all traits, i.e. HCM + LV traits but rather provides improved estimates of HCM~SNP effects by incorporating information of highly correlated LV traits. Practically, MTAG combining HCM + 3 LV traits yields summary statistics for each of the 4 included traits (HCM and each of the 3 LV traits). For the purpose of the present study, we only use MTAG results for HCM, and not MTAG of the other included traits.

The robustness of the MTAG approach in HCM has been discussed in our responses to the other reviewers. Importantly, our prior publication (Tadros et al PMID 33495596) had identified 10 novel HCM loci using MTAG. All those new MTAG loci were successfully validated in an independent cohort.

Furthermore, all those 10 MTAG loci now reach significance ($P < 5 \times 10^{-8}$) in the present GWAS. This highlights the robustness of this approach for discovery of HCM-associated loci.

With regard to the ancestral composition of the case and control sets, the HCM cases were enrolled in a number of countries but the majority of the controls are from the UK. A sensitivity analysis restricted to only UK ancestry might determine if the signals are driven entirely by UK ancestry.

The genome-wide association analyses presented consist of a meta-analysis of 7 case-control HCM datasets, each with a control dataset that is ancestrally matched with the cases. This includes a Canadian dataset comprised of 1,035 Canadian HCM cases and 13,889 Canadian controls, a Dutch dataset comprised of 999 Dutch HCM cases and 2,117 Dutch HCM controls, an Italian dataset comprised of 277 Italian HCM cases and 1,293 Italian controls, and 3 UK case-control datasets from the Royal Brompton, BRRD and GEL. The last case-control dataset is comprised of cases from the Trans-Atlantic HCM registry (HCMR) and controls from the UK Biobank. Although this last case-control dataset which was previously published (Harper et al. PMID 33495597) may hypothetically introduce a population stratification bias, the genomic inflation factor (λ_{GC}) was appropriately small (1.05) following correction for residual ancestral differences captured with genotypic principal components. More importantly, we now show meta-analysis heterogeneity metrics including Cochrane's Q heterogeneity test and forest plots for all GWAS and MTAG lead variants (Table 1, Supplementary Table 8, Supplementary Figures 1 and 14). These metrics do not suggest that significant loci are "driven entirely" by the UK controls of the HCMR dataset.

Conclusions: robustness, validity, reliability

Overall, I feel that the findings, particularly in relation to the 'new gene' are somewhat oversold and a bit forced into the paper.

In addition to extending upon the rare variant association analyses, including adding further data in support of *SVIL*, we have toned down the wording regarding *SVIL* association with HCM, in the rare variant association paragraph.

NG-LE61789R1

Large scale genome-wide association analyses identify novel genetic loci and mechanisms in hypertrophic cardiomyopathy

Response to reviewers' comments

We thank the Editors and Reviewers for the careful review and for providing us with an opportunity to improve our manuscript once again. We hereby provide a point-by-point response in red to each Reviewer comment (copied in black). Changes in the manuscript or supporting tables and figures, if any, are described within each response. When appropriate, the text change is also copied in the response. All changes made are highlighted in the main manuscript text.

Editors' prioritized set of referee points:

Extend the rare variant burden analyses:

- a) to include all genes
- b) to explore results obtained using a less stringent minor allele frequency filter
- c) to present the corresponding QQ plots for the rare variant burden analyses
- d) to make the full summary statistics available

We thank the editors for providing a prioritized set of referee points, which we address within our response to the reviewers.

Reviewer #1:

Thank you to the authors for their extensive work to respond to my remarks and the other reviewers. I am satisfied with the responses to my questions and the additional information added to the paper, new figure 4 works well and great to have additional reporting on results from the MTAG approach. The inclusion of testing for rare LoF variants across 26 candidate genes is a good addition to the paper. My only remaining comment concerns you may wish to revise the abstract to demonstrate additional work done on rare variant testing on candidate genes?

We thank the reviewer for this suggestion. We have now modified the corresponding sentence in the abstract, as follows: "Amongst the prioritized genes in the HCM loci, we identify a novel HCM disease gene, *SVIL*, which encodes the actin-binding protein supervillin, showing that rare truncating *SVIL* variants cause HCM"

Reviewer #2:

The authors' response has addressed many of the reviewers' concerns. There are still some remaining questions that were not addressed.

1) The authors extended the burden analysis to 26 genes rather than looking at the whole exome to maintain statistical power. I do not think this is sufficient and I think the authors should test all genes for associations in a burden test. They can still test the 26 genes in a more focused hypothesis and

adjust only for the 26 genes tested. It is important that the authors provide QQ plots for the burden analysis for quality control and I cannot see any reason for not doing so. The negative control analysis is not sufficient. Furthermore, I think the authors should make the summary level data from the burden analysis publicly available to other researchers as is now standard to do with GWAS data.

As suggested by the reviewer, we have now performed gene-based burden testing for the whole exome as an exploratory analysis, using 2 MAF thresholds (<0.0001 and <0.001). This is described in new sections of text in results on page 7 and methods on page 25. We provide QQ plots at cohort-level (**Supplementary Figure 16**) and at meta-analysis level (**Supplementary Figure 17**, reproduced below). We also provide Manhattan plots for the burden analyses (**Supplementary Figure 18**, reproduced below). Last, as requested by the reviewer and editors, we provide the full summary level results in **Supplementary Tables 17-18**.

In the inverse variance weighted (IVW) meta-analysis, we observed some statistical inflation. This likely reflects the exclusion of gene/datasets where there are no case carriers of rare loss of function variants, because of inability to compute a standard error in such cases with 0 count. This results in preferential inclusion of genes where there is possible enrichment of loss-of-function variants in cases, therefore violating the null hypothesis and resulting in statistical inflation in the IVW meta-analysis. As a more conservative approach, we also performed sample size weighted (SSW) meta-analysis and report such results along with the IVW meta-analysis (**Supplementary Tables 16-18**). The SSW meta-analysis allows for inclusion of gene/datasets with 0 count (where a P value can still be computed, but not a standard error) and may therefore be less biased, as shown on the QQ plot below.

Reproduced **Supplementary Figure 17**

As expected from the limited statistical power, few genes reach the Bonferroni-corrected threshold for significance in this exome-wide exploratory analysis ($P = 0.05/16,508 = 3 \times 10^{-6}$), with only *MYBPC3* and *ALPK3* reaching such threshold consistently across all 4 analyses (IVW and SSW, using $MAF < 0.0001$ and $MAF < 0.001$). Association of loss of function variants in *SVIL* with HCM reached the exploratory exome-wide P-value threshold in the IVW meta-analysis using both MAF thresholds. It also reached the primary targeted analysis of prioritized genes threshold of $0.05/26$ (0.0019) in all 4 meta-analyses (P values ranging from 0.0000004 to 0.0001 ; see **Supplementary Tables 16-18**). Given inconsistency across the IVW (inflated statistics) and SSW meta-analyses and across the 2 MAF thresholds, and considering the exploratory nature of the exome-wide analyses, we believe that other loci reaching the $P < 3 \times 10^{-6}$ threshold likely represent false positive signals.

Reproduced **Supplementary Figure 18**

2) The authors claim that it is standard and necessary to use a $MAF < 10^{-4}$ filter in the analysis and state that no HCM above that frequency has been reported. I do not agree with that. There are e.g. founder mutations in *MYBPC3* that have a much higher frequency and cause HCM. I think all predicted loss-of-function variants in the genes should be explored. If there are more frequent variants, e.g. in *SVIL*, that do not associate with HCM and render the burden association non-significant, that should then be discussed.

As suggested by the reviewer and the editors, we also explored burden testing results using a less stringent minor allele frequency (MAF) filter ($MAF < 10^{-3}$). The association of loss-of-function variants in *SVIL* with HCM remains significant (OR 9.6; $P = 5 \times 10^{-6}$), as is the case for *MYBPC3* and *ALPK3*. Note that there are no predicted loss of function variant in *SVIL* with $MAF > 10^{-4}$ in gnomAD v4.

3) The authors state that exploring a bi-directional relationship between HCM and LV contractility is outside of the scope of the manuscript. Confounding due to reverse causation, however, can complicate the interpretation of MR findings. I feel that it needs to be explored and potentially bi-directional relationships need to be accounted for in MR analysis.

The reviewer is correct that MR findings always need to be interpreted with caution. As mentioned in the manuscript, the MR findings are *supportive* of a causal relationship but are not conclusive proof of causal relationship in the context of potential heterogeneity and possible violation of MR assumptions. Further data from ongoing clinical trials are needed to confirm such a causal relationship. Assessing directionality in MR can be complex when one of the exposure/outcome pair is binary (as is the case for HCM). Given such complexity, the lack of a biological rationale for reverse causation, and considering that the editors did not include this in the prioritized set of referee's points, we did not perform or report bi-directional MR.